# PHC1 maintains pluripotency by organizing genome-wide chromatin interactions of the *Nanog* locus

Li Chen[1,2,3,14], Qiaoqiao Tong[1,2,3,14], Xiaowen Chen[4,14], Penglei Jiang[1,2], Hua Yu[1,2], Qianbing Zhao[1,2,3], Lingang Sun[1,2,3], Chao Liu[1,2,3,5], Bin Gu[6], Yuping Zheng[7], Lijiang Fei[1,2,3], Xiao Jiang[8], Wenjuan Li[9,10], Giacomo Volpe[9,10], Mazid MD. Abdul[9,10], Guoji Guo[1,2,3], Jin Zhang[1,2], Pengxu Qian[1,2], Qiming Sun[8], Dante Neculai[7], Miguel A. Esteban[9,10,11], Chen Li[12✉], Feiqiu Wen[4✉] & Junfeng Ji[1,2,3,13✉]

Polycomb group (PcG) proteins maintain cell identity by repressing gene expression during development. Surprisingly, emerging studies have recently reported that a number of PcG proteins directly activate gene expression during cell fate determination process. However, the mechanisms by which they direct gene activation in pluripotency remain poorly understood. Here, we show that Phc1, a subunit of canonical polycomb repressive complex 1 (cPRC1), can exert its function in pluripotency maintenance via a PRC1-independent activation of *Nanog*. Ablation of *Phc1* reduces the expression of *Nanog* and overexpression of *Nanog* partially rescues impaired pluripotency caused by *Phc1* depletion. We find that Phc1 interacts with Nanog and activates *Nanog* transcription by stabilizing the genome-wide chromatin interactions of the *Nanog* locus. This adds to the already known canonical function of PRC1 in pluripotency maintenance via a PRC1-dependent repression of differentiation genes. Overall, our study reveals a function of Phc1 to activate *Nanog* transcription through regulating chromatin architecture and proposes a paradigm for PcG proteins to maintain pluripotency.

[1] Center of Stem Cell and Regenerative Medicine, and Bone Marrow Transplantation Center of the First Affiliated Hospital, Zhejiang University School of Medicine, Hangzhou 310058, China. [2] Zhejiang Engineering Laboratory for Stem Cell and Immunotherapy, Institute of Hematology, Zhejiang University, Hangzhou 310058, China. [3] Zhejiang Provincial Key Laboratory of Tissue Engineering and Regenerative Medicine, Hangzhou 310058, China. [4] Division of Hematology and Oncology, Shenzhen Children's Hospital, Shenzhen 518026, China. [5] ZJU-UoE Joint Institute, School of Medicine, Zhejiang University, Hangzhou 310058, China. [6] Program in Developmental and Stem Cell Biology, Hospital for Sick Children, Toronto, Canada. [7] Department of Cell Biology, School of Medicine, Zhejiang University, Hangzhou 310058, China. [8] Department of Biochemistry, Department of Cardiology of Second Affiliated Hospital, Zhejiang University School of Medicine, Hangzhou 310058, China. [9] Laboratory of Integrative Biology, Guangzhou Institutes of Biomedicine and Health, Chinese Academy of Sciences, Guangzhou 510530, China. [10] Key Laboratory of Regenerative Biology of the Chinese Academy of Sciences and Guangdong Provincial Key Laboratory of Stem Cells and Regenerative Medicine, Guangzhou Institutes of Biomedicine and Health, Chinese Academy of Sciences, Guangzhou 510530, China. [11] Bioland Laboratory (Guangzhou Regenerative Medicine and Health Guangdong Laboratory), Guangzhou 510005, China. [12] Department of Human Genetics, and Women's Hospital, Zhejiang University School of Medicine, Hangzhou 310058, China. [13] Zhejiang Laboratory for Systems & Precision Medicine, Zhejiang University Medical Center, Hangzhou 311121, China. [14] These authors contributed equally: Li Chen, Qiaoqiao Tong, Xiaowen Chen. ✉email: chenli2012@zju.edu.cn; fwen62@126.com; Jijunfeng@zju.edu.cn

Pluripotent stem cells (PSCs) can self-renew indefinitely in culture while maintaining the potency to give rise to any cell types of the three germ layers[1,2]. Master transcription factors (TFs) including Oct4, Sox2, and Nanog (OSN) are essential for the maintenance of pluripotency by forming the core autoregulatory circuitry that orchestrates a pluripotent-specific transcriptional network[3–6]. At three-dimensional (3D) genome level, recent studies have shown that OSN together with coactivators such as mediator and the transcriptional apparatus bind to cis-regulatory elements termed super-enhancers (SEs) of gene loci associated with pluripotent cell identity[7,8]. SEs which are defined by high OSN occupancy and enrichment of active chromatin marks such as histone 3 lysine 27 acetylation (H3K27ac) in PSCs, contact multiple promoters to form pluripotency-specific interaction networks[9,10]. Therefore, master TFs cooperate with epigenetic modifiers to drive robust expression of pluripotent genes by shaping the chromatin landscape[9].

Polycomb group (PcG) proteins maintain cell identity by transcriptionally silencing developmental genes during embryogenesis[10,11]. PcG proteins assemble into two major multi-subunit complexes called the polycomb repressive complex 1 and 2 (PRC1 and PRC2). PRC2 catalyzes tri-methylation of histone H3 at lysine 27 (H3K27me3) through the histone methyltransferase Ezh1/2[12]. The PRC1 complexes contain the E3 ligases Ring1A/B and one of Pcgf proteins as the core, capable of mono-ubiquitinating lysine 119 of histone H2A (H2AK119ub1)[11]. Based on the subunit composition, PRC1 is further classified into canonical or non-canonical PRC1 (cPRC1 and ncPRC1). The Cbx subunit (Cbx2/4/6-8) of the cPRC1 complex reads PRC2-mediated H3K27me3 and recruits it to the chromatin, and the Phc subunit (Phc1-3) promotes subsequent chromatin condensation and gene silencing through oligomerization of the sterile alpha motif (SAM)[11]. However, the ncPRC1 complex contains a Rybp or Raf2 subunit, but lacks both Phc and Cbx proteins, and adopts a PRC2/H3K27me3-independent targeting to the chromatin which drives PRC2 recruitment[13–15]. Thus, PRC2 and PRC1 act concertedly to deposit H3K27me3 and H2AK119ub1 histone modifications, thereby establishing a repressive chromatin landscape to silence gene expression[11]. In PSCs, both PRC2 and PRC1 bind to developmental regulators and repress their expression thereby maintaining the cells in the undifferentiated state[16,17]. Recent Hi-C (high-throughput Chromosome Conformation Capture) technology has shown that PRC1-bound sites in the genome contact each other to form chromatin loops through PHC-SAM polymerization for developmental gene repression[18–22]. Intriguingly, besides their well-established gene repression function, several studies have reported that a number of PcG proteins, PRC1 subunits in particular, directly activate gene expression to regulate cell fate decision in a PRC1-independent manner[23–31]. In PSCs, PRC1 proteins such as Rybp and Pcgf5 have been reported to associate with TFs to activate transcription[26,28]. However, the molecular mechanisms underpinning how PcG proteins cooperate with master TFs to activate pluripotency-associated genes remain poorly understood.

In this work, we show that Phc1 is highly enriched in pluripotent cells in comparison with differentiated cells, and depletion of Phc1 in both mouse and human embryonic stem cells (mESCs and hESCs) compromises pluripotency partially through reducing Nanog expression. Furthermore, Phc1 interacts with Nanog and activates its transcription by stabilizing intra- and inter-chromosomal interactions of the Nanog locus. Our study demonstrates that Phc1 maintains pluripotency not only through PRC1-dependent repression of developmental genes, but also through organizing genome-wide chromatin interactions of the Nanog locus.

## Results

**PHC1 is highly expressed in pluripotent cells.** In order to identify cPRC1 genes that are highly expressed in human pluripotent stem cells (hPSCs), we performed differential gene expression analysis of all the cPRC1 subunits as well as core PRC2 subunits. We observed highly enriched expression of CBX2, CBX7, PHC1, PHC3, RING1B, and EZH2 in hESCs (H9) and human induced pluripotent stem cells (iPSCs) in comparison to human foreskin fibroblasts (HFFs) (Fig. 1a). In particular, we found that the expression of PHC1, similar to key pluripotency factors POU5F1 and NANOG, declined during differentiation of embryoid bodies (EBs), indicating that PHC1 was specifically expressed in the undifferentiated PSCs (Fig. 1b). This finding was further confirmed by western blot (WB) analysis in multiple hPSCs including hESCs, iPSCs, and NCCIT, a human teratocarcinoma cell line (Fig. 1c). Next, we analyzed the publicly available single-cell RNA sequencing data of early human embryos and found that the expression of PHC1 among other analyzed PRC1 genes such as RING1B, PHC2, PHC3, and CBX2 was enriched in epiblast (EPI) relative to trophectoderm (TE) and primitive endoderm (PE) lineages (Fig. 1d and Supplementary Fig. 1a)[32]. Moreover, the expression of NANOG is positively correlated with that of PHC1, but not RING1B, at E5 EPI lineage (Fig. 1d and Supplementary Fig. 1b). Immunostaining of E4.5 mouse embryos demonstrated that the expression of Phc1 co-localized with Nanog and Sox2 in EPI, but not with PE marker Gata6 (Fig. 1e). Taken together, these results demonstrate that PHC1 is highly enriched in both human and mouse pluripotent cells as opposed to the differentiated cells therefore suggesting PHC1 may play an important role in the maintenance of pluripotency.

**PHC1 maintains pluripotency of hESCs beyond PRC1.** To functionally evaluate the role of PHC1 in the maintenance of hESCs, we performed shRNA-mediated knockdown using shRNAs targeting either the human PHC1 gene (shPHC1.1 & shPHC1.2) or a scrambled (shScr) negative control (Supplementary Fig. 2a, b and Supplementary Table 3). Suppression of PHC1 expression by both shRNAs induced differentiation and significantly decreased the colony-forming capacity of hESCs (Fig. 2a, b), which indicates that PHC1 is required for the self-renewal of hESCs. We further performed teratoma formation assay in NOD/SCID mice. Knocking down PHC1 in particular with shPHC1.1 significantly reduced the tumor sizes compared to that of the control (Fig. 2c). Histological analysis of the tumors showed that, in comparison with the control, hESCs after PHC1 suppression gave rise to less mature neural, gland, and muscle tissues representative of ectoderm, endoderm, and mesoderm, respectively (Supplementary Fig. 2c). These results demonstrated that PHC1 is required for the pluripotency maintenance of hESCs in vivo. At the molecular level, knocking down PHC1 decreased the level of NANOG, but not that of OCT4 and SOX2 (Fig. 2d). Interestingly, suppression of PHC1 expression did not obviously change the levels of RING1B, the E3 ligase subunit of PRC1, and its associated histone modification H2AK119ub1 level was not consistently affected (Fig. 2d). This result implies that PHC1 knockdown impairs pluripotency of hESCs independently of PRC1. To further verify that PHC1 suppression downregulates NANOG level, we employed a clustered regulatory interspaced short palindromic repeats (CRISPR)-associated protein 9 (Cas9) approach to knockout human PHC1 gene in hESCs by targeting exon 7 (Fig. 2e). This led to drastic downregulation of NANOG expression concomitant with PHC1 ablation in the bulk population of hESCs (Fig. 2e). This observation is consistent with the shRNA-based knockdown results and indicates that

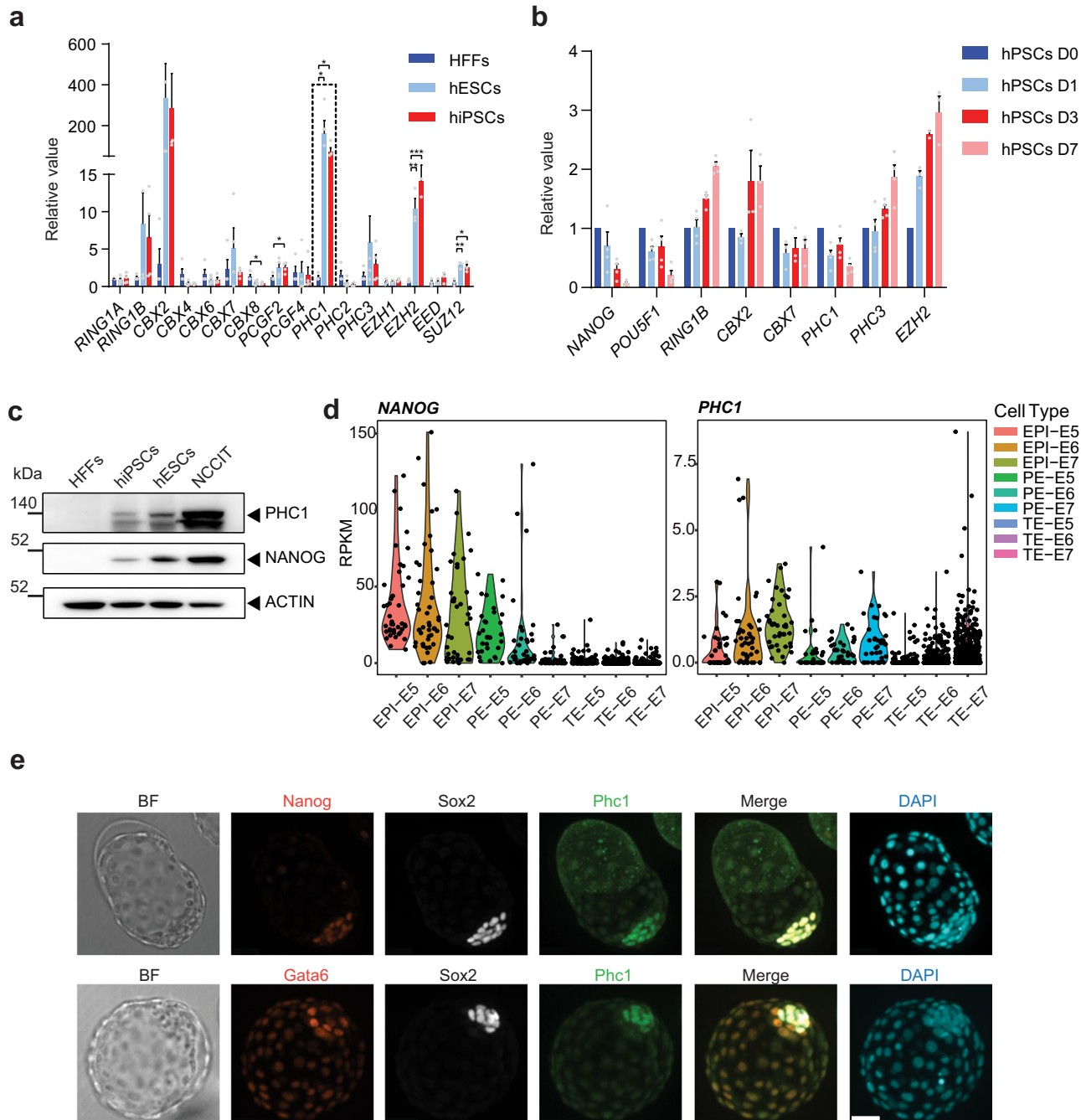

**Fig. 1 PHC1 is highly expressed in both human and mouse pluripotent cells. a** PRC1 and PRC2 gene expression was examined by RT-qPCR in HFFs, hESCs, and hiPSCs. *ACTIN* was used as the house keeping gene and expression was normalized to HFFs. $n = 4$ independent experiments for PRC1 genes; $n = 3$ independent experiments for PRC2 genes including *EZH1, EZH2, EED,* and *SUZ12*. Error bars represent the s.e.m. Two-tailed unpaired *t*-tests were used ($p = 0.0256$ for *CBX8*, $p = 0.0201$ for *PCGF2*, $p = 0.0498$ for *PHC1* in HFFs vs. ESCs, $p = 0.0104$ for *PHC1* in HFFs vs. iPSCs, **$p = 0.0022$ for *EZH2*, ***$p = 0.0003$ for *EZH2*, **$p = 0.0034$ for *SUZ12*, *$p = 0.0111$ for *SUZ12*). **b** Expressions of PRC1 and PRC2 genes and key pluripotency factors *POU5F1* and *NANOG* during embryoid body differentiation (D0–D7) of hPSCs were examined by RT-qPCR. $n = 3$ independent experiments for *CBX2, CBX7,* and *EZH2*; $n = 4$ independent experiments for the other genes. Error bars represent the s.e.m. **c** WB analysis of PHC1 and NANOG protein expression in hPSCs and HFFs. **d** Analysis of the published single-cell RNA-seq data of early human embryos (E5–7) showing expression of *NANOG* and *PHC1* in epiblast (Epi), primitive endoderm (PE), and trophectoderm (TE)[32]. **e** Co-immunostaining of Phc1 with Nanog and Sox2, or Gata6 and Sox2 in mouse E4.5 embryos. Scale bars, 31 μm. Source data are provided as a Source Data file.

downregulation of NANOG caused by *PHC1* suppression was unlikely due to an off-target effect (Fig. 2d). Furthermore, immunofluorescent co-staining of PHC1 with NANOG, OCT4, or SOX2 in hESCs also showed that, at single-cell level *PHC1* suppression correlated with lower expression of NANOG, but not OCT4 and SOX2 (Fig. 2f). The percentage of

PHC1^lowNANOG^low cells was significantly higher than that of PHC1^highNANOG^low cells due to heterogeneous expression of NANOG in the *shPHC1*-infected hESC population as previously reported[33] (Fig. 2g). This result also supports that *PHC1* suppression specifically reduces NANOG expression (Fig. 2g). Taken together, these results imply that PHC1 is required for the

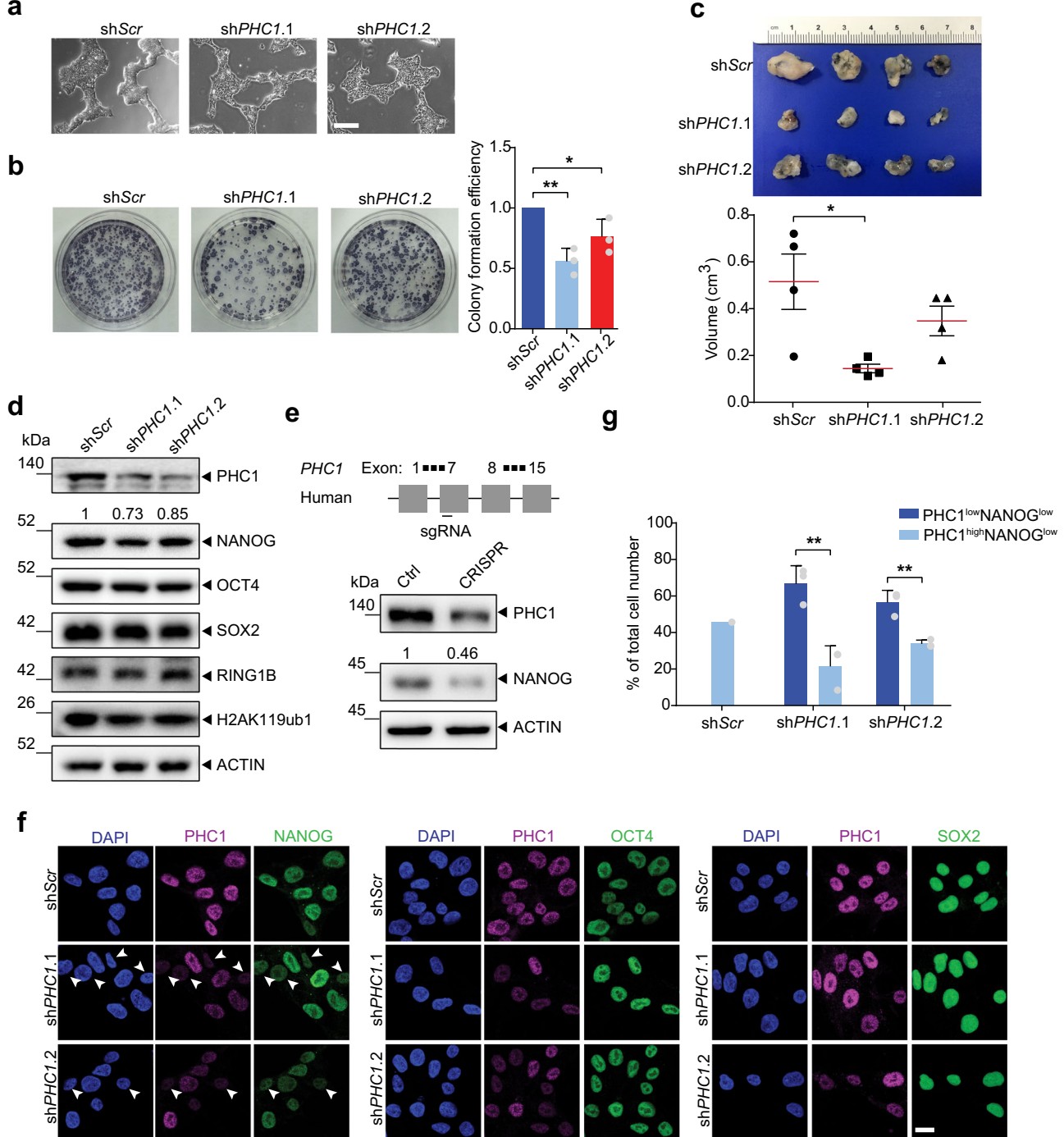

**Fig. 2 PHC1 is required for the pluripotency maintenance of hESCs. a**, **b** Morphology and colony-forming capacity of sh*Scr*-, sh*PHC1*.1-, and sh*PHC1*.2-infected hESCs. Scale bars, 600 μm. Cells were stained for alkaline phosphatase activity 14 or 18 days after plating. Bar plot shows mean colony-formation efficiencies normalized to the sh*Scr*. Mean ± s.d. of $n = 3$ independent experiments. Two-tailed unpaired $t$-tests were used (**$p = 0.0022$, *$p = 0.0458$). **c** Comparison of tumor sizes about 1 month after injection of sh*Scr*-, and sh*PHC1*-infected hESCs into NOD/SCID mice. Data are presented as mean ± s.e.m. In each group, 4 NOD/SCID mice were injected. Two-tailed unpaired $t$-tests were used (*$p = 0.0211$). **d** WB analysis of PHC1, NANOG, OCT4, SOX2, RING1B, H2AK119ub1, and ACTIN protein levels in sh*Scr* and sh*PHC1*-infected hESCs. **e** WB analysis of PHC1 and NANOG protein levels in hESCs infected with control and CRISPR-CAS9 vector targeting human *PHC1* gene. **f** Immunofluorescent co-staining of PHC1 with NANOG, OCT4, or SOX2 in sh*Scr*- and sh*PHC1*-infected hESCs. Arrowheads showed cells with low PHC1 and NANOG signals. Scale bars, 20 μm. **g** The ratios of cells exhibiting PHC1$^{high}$NANOG$^{low}$ and PHC1$^{low}$NANOG$^{low}$ signals to the total number of stained cells in (**f**) were quantified. Mean ± s.d. of $n = 3$ independent counts for sh*PHC1*.1 and sh*PHC1*.2. Two-tailed unpaired $t$-tests were used ($p = 0.0065$ for sh*PHC1*.1, $p = 0.0048$ for sh*PHC1*.2). Source data are provided as a Source Data file.

maintenance of hESCs possibly through specifically regulating *NANOG* expression.

Since PHC1 is a subunit of cPRC1, we looked into the impact of *PHC1* suppression on PRC1 in hESCs. We first examined the composition of PRC1 in hESCs. Super-resolution microscopic imaging analysis of PHC1 and RING1B co-immunostaining showed that, while the majority (around 80%) of PHC1 and RING1B signals overlapped, some did not co-localize with each other (Supplementary Fig. 3a, b). This observation indicates that PHC1 and RING1B associate primarily, but not exclusively, to form cPRC1 in hESCs. Interestingly, despite RING1B being known to associate with various subunits to form ncPRC1 in the absence of PHC1[11], RING1B-nonoverlapping PHC1 (approximately 20%) signals suggest that PHC1 may possess PRC1-independent functions. To dissect this, we analyzed the publicly available Phc1 and Ring1B ChIP-seq data in mESCs[16]. We observed that while Phc1 and Ring1b co-occupied majority of their bound targets (Phc1: $1095/1734 \approx 63\%$; Ring1b: $1095/6349 \approx 17\%$), there were still approximately 24% of Phc1-bound genes and 43% of Ring 1b occupied targets that did not overlap with each other (Supplementary Fig. 3c). *Ano2* and *Foxj2* genes with nonoverlapping Phc1 and Ring1b binding peaks were shown as the examples (Supplementary Fig. 3d). Furthermore, by performing immunoprecipitation (IP) experiments, we showed that endogenous RING1B was able to precipitate both PHC1 and CBX7, subunits of cPRC1, and RYBP, a core constituent of ncPRC1 (Supplementary Fig. 3e). This result indicates that RING1B associates with PHC1 and CBX7 as well as RYBP to form cPRC1 and ncPRC1 in hESCs, respectively. Further IP results showed that *PHC1* suppression neither significantly affected the expressions of CBX7, RYBP, and RING1B, nor their precipitations by RING1B (Supplementary Fig. 3f). This finding indicates that *PHC1* knockdown did not disrupt the assembly of PRC1 in hESCs. We then asked whether *PHC1* knockdown affected the occupancy of PRC1 target genes in hESCs by performing H2AK119ub1 ChIP-PCR (Supplementary Figs. 3g and 4). Suppression of *PHC1* did not significantly change the H2AK119ub1 enrichment levels of *EOMES*, *GSX2*, *EX2*, *GATA4*, and *GATA6* (Supplementary Fig. 3g), known PRC1 target developmental genes representative of mesoderm, ectoderm, and primitive endoderm, respectively[17]. Thus, it demonstrates that *PHC1* knockdown did not influence PRC1-associated H2AK119ub1 enrichment of target differentiation genes in hESCs. Taken together, these results indicate that PHC1 may possess a PRC1-independent role in maintaining pluripotency of hESCs.

**PRC1-independent regulation of *Nanog* by Phc1 in mESCs.** To further investigate the mechanisms by which PHC1 regulates pluripotency, we used CRISPR-Cas9 method to completely knockout *Phc1* in mESCs by two pairs of guide RNAs independently targeting exon 2 and exons 7–11, respectively (Fig. 3a, Supplementary Fig. 5a, and Supplementary Table 3). This led to the generation of four *Phc1*-deficient ($Phc1^{-/-}$) mESC clones verified by WB analysis which showed the absence of Phc1 protein (Fig. 3a, b and Supplementary Fig. 5a, b). Depletion of *Phc1* induced spontaneous differentiation of mESCs (Fig. 3a and Supplementary Fig. 5a). Consistent with *PHC1* knockdown results in hESCs (Fig. 2d, e), *Phc1* knockout in mESCs significantly reduced the expression of Nanog protein in multiple clones (Fig. 3b and Supplementary Fig. 5b), whereas Oct4 and Sox2 expression levels did not significantly change (Fig. 3b). By contrast, knocking down *Ring1b* did not significantly affect Nanog expression despite that it caused derepression of a few PRC1 target genes (Supplementary Fig. 5c, d). This result

demonstrates that Phc1 specifically regulates Nanog among the core pluripotent TFs. qPCR analysis showed that ablation of *Phc1* led to significant reduction in the transcript abundance of *Nanog* and its known direct target genes such as *Klf4* and *Esrrb* (Fig. 3c)[34,35], indicating that Phc1 positively regulates the transcription of *Nanog*. Furthermore, we knocked down *Phc1* in a mESC line carrying the *Nanog*-GFP reporter. Flow cytometric analysis showed that suppression of *Phc1* significantly reduced the fraction of GFP⁺ cells similar to *Nanog* knockdown positive control (Fig. 3d), further supporting the idea of Phc1 regulating *Nanog* at the transcriptional level. To examine if reduced Nanog expression in part mediates the effect of *Phc1* depletion on pluripotency maintenance of mESCs, we overexpressed exogenous Nanog tagged with Flag epitope in the wild-type and $Phc1^{-/-}$ mESC line. In line with the role of Nanog in supporting self-renewal of mESCs[36], overexpression of exogenous Nanog in the $Phc1^{-/-}$ mESCs restored their undifferentiated morphology, clonogenicity, and expressions of its target genes associated with pluripotency and PE, but not the non-target genes involved in mesoderm and ectoderm differentiation (Fig. 3e–h and Supplementary Fig. 5e). Taken together, our results demonstrated that Phc1 functions upstream of *Nanog* and maintains pluripotency of mESCs at least in part through positively regulating *Nanog*.

To discern between the PRC1-dependent and -independent function of Phc1 in maintaining the pluripotency of mESCs, we analyzed the available ChIP-seq data of Phc1 against that of PRC1-dependent repressive H2AK119ub1 and PRC1-independent active H3K27ac chromatin marks[22,37]. The analysis showed that approximately 51% (884/1734) of the binding targets of Phc1 overlapped with that of H2AK119ub1 (Fig. 4a) and as expected many of the co-occupied targets are development-associated genes as shown by GO pathway analysis (Fig. 4b)[16,22]. Importantly, around 72% (1251/1734) of Phc1 target genes including genes that are involved in regulating pluripotency of stem cells were also occupied by active chromatin mark H3K27ac, pointing out that Phc1 could also activate gene expression to maintain pluripotency of mESCs (Fig. 4a, b). To fully understand how Phc1 regulates the pluripotency of mESCs in both PRC1-dependent and -independent manners, we performed RNA sequencing of the $Phc1^{+/+}$ mESCs + vector, $Phc1^{-/-}$ mESCs + vector, and $Phc1^{-/-}$ mESCs + Nanog overexpression followed by analysis against Phc1, H2AK119ub1, Nanog, and H3K27ac ChIP-seq datasets (Fig. 4c). The results showed that depletion of *Phc1* induced downregulation of pluripotency-associated genes including *Nanog*, *Esrrb*, and *Satb1*, etc. (Fig. 4c yellow and blue clusters, and Supplementary Data 1). In contrast, overexpression of Nanog in the $Phc1^{-/-}$ mESCs partly rescued the expression of a subset of genes including *Esrrb* which were bound by both Nanog and H3K27ac, but devoid of Phc1 binding and H2Ak119ub1 mark (Fig. 4c blue cluster, and Supplementary Data 1) as previously shown[34]. However, the expression levels of some pluripotency-associated genes such as *Satb1* were not restored by overexpression of Nanog (Fig. 4c yellow cluster, and Supplementary Data 1). Thus, these results demonstrated that Phc1 activates pluripotency-related genes in both Nanog-dependent and -independent manner. Moreover, consistent with a previous study[22], ablation of *Phc1* also upregulated various lineage-differentiation genes including *Hox* and neural development-associated genes which were co-occupied by Phc1 and H2AK119ub1 (Fig. 4c purple cluster, and Supplementary Data 1), supporting that Phc1 repressed the expression of these genes in a PRC1-dependent manner. Overexpression of Nanog in $Phc1^{-/-}$ mESCs reduced their expression levels similar to that of $Phc1^{+/+}$ mESCs + vector (Fig. 4c purple cluster, and Supplementary Data 1), indicating that Phc1 represses this cluster of differentiation-associated genes in a Nanog-dependent manner.

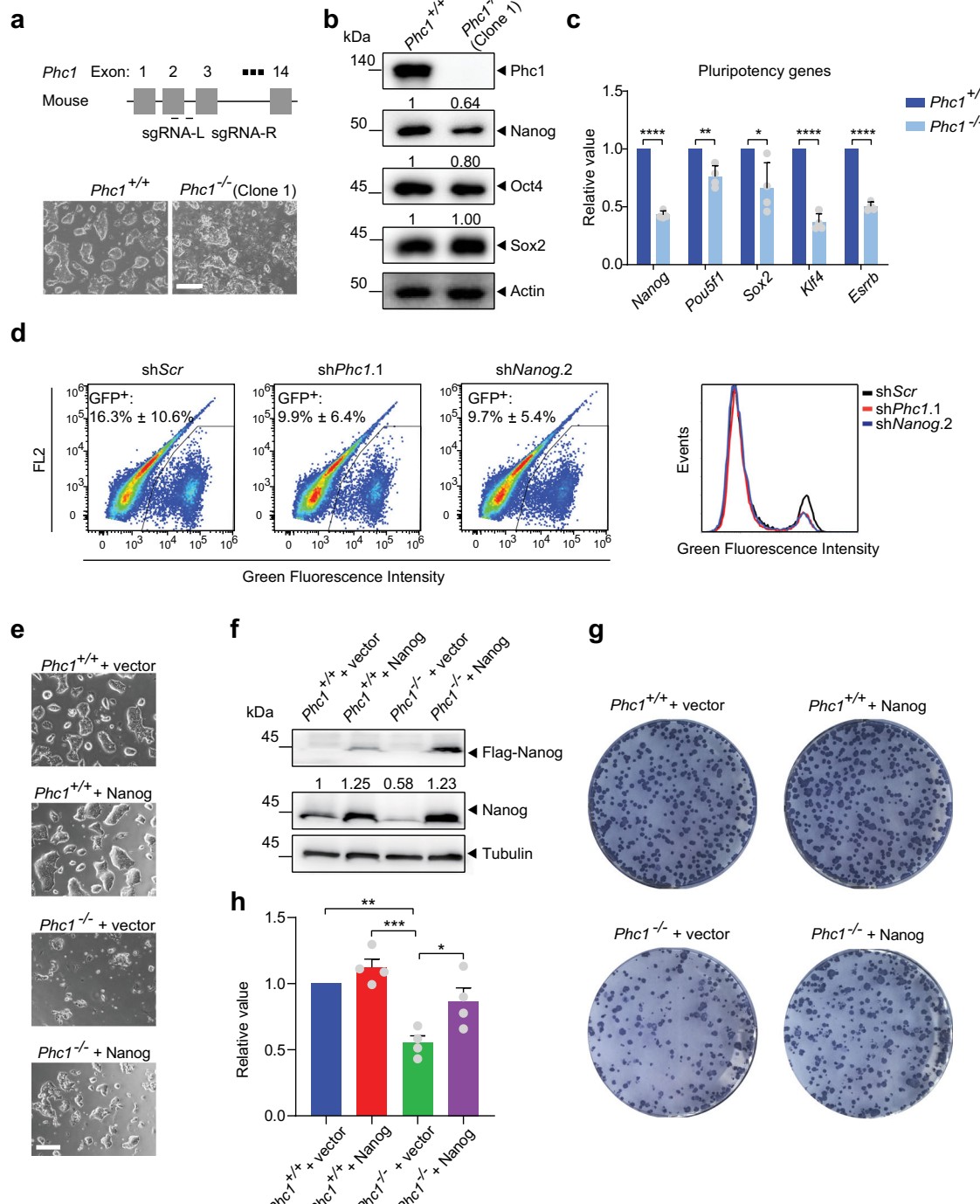

**Fig. 3 Phc1 maintains pluripotency of mESCs partly through Nanog. a** Designing sgRNAs targeting the 2nd exon and 3rd intron of mouse *Phc1*, and the morphology of the *Phc1*+/+ and *Phc1*−/− mESCs. Scale bars, 300 μm. **b** WB analysis of Phc1, Nanog, Oct4, Sox2, and Actin protein levels in the *Phc1*+/+ and *Phc1*−/− mESCs. **c** qPCR analysis of transcript levels of *Pou5f1, Sox2 Nanog*, and its known direct target genes including *Klf4* and *Esrrb*. Mean ± s.d. of $n = 4$ independent experiments. Two-tailed *t*-tests were used ($p < 0.0001$ for *Nanog*; $p = 0.0023$ for *Pou5f1*; $p = 0.0216$ for *Sox2*; $p < 0.0001$ for *Klf4*; $p < 0.0001$ for *Esrrb*). **d** Flow cytometry analysis quantifying GFP signal after *Phc1* knockdown in a mESC line carrying the *Nanog*-GFP reporter. *Nanog* suppression was used as the positive control. Mean ± s.d. of $n = 3$ independent experiments. **e** Morphology of *Phc1*+/+ and *Phc1*−/− mESCs transfected with an empty or Flag-Nanog vector. Scale bars, 600 μm. **f** Immunoblotting of Flag, Nanog, and Tubulin in extracts of mESCs in (**e**). **g** Alkaline phosphatase staining of mESCs in (**e**). **h** Quantification of colony-forming efficiencies of mESCs relative to *Phc1*+/+ + vector as the control in (**e**). Mean ± s.e.m. of $n = 4$ independent experiments. One-way ANOVA test with Bonferroni's multiple comparison was used (**$p = 0.0024$, ***$p = 0.0003$, *$p = 0.0319$). Source data are provided as a Source Data file.

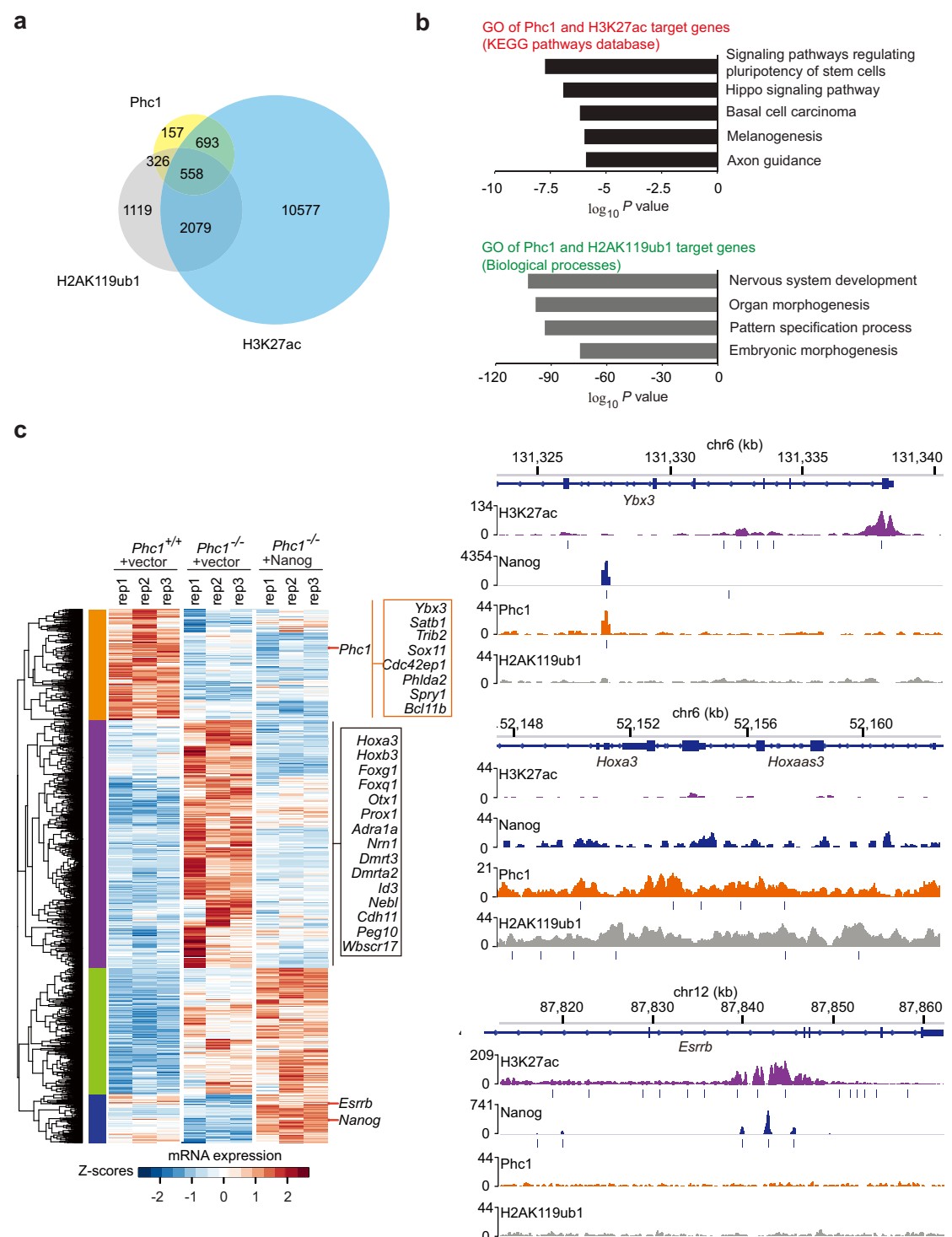

**Fig. 4 Phc1 maintains pluripotency of mESCs in both PRC1-dependent and PRC1-independent regulation of Nanog. a** Venn diagram analysis of genes bound by Phc1, H2AK119ub1, and H3K27ac in mESCs[22,37]. **b** GO analysis of signaling pathways of Phc1 and H3K27ac target genes and biological functions of genes co-occupied by Phc1 and H2AK119ub1. $p$ values are plotted in –log[10]. **c** The $Phc1^{+/+}$ + vector, $Phc1^{-/-}$ + vector, and $Phc1^{-/-}$ + Nanog mESCs were profiled in triplicate for RNA-seq experiments. Genes that were significantly up- or down-regulated in $Phc1^{-/-}$ + vector compared with the $Phc1^{+/+}$ + vector cells were clustered across all samples and were shown as heatmaps. Each row represents one gene and each column represents one sample. The Phc1-activated genes that were not changed by Nanog overexpression were shown as orange cluster and that were induced by Nanog were classified as blue cluster. Genes that were repressed by both Phc1 and Nanog were defined as purple cluster. The Phc1-repressed genes but not repressed by Nanog were shown as green cluster. The IGV image showing H3K27ac, Nanog, Phc1, and H2AK119ub1 ChIP-seq binding profiles of representative genes from orange and purple cluster genes[8,22,37]. Source data are provided as a Source Data file.

In contrast, some upregulated genes in the $Phc1^{-/-}$ mESCs were not rescued by overexpression of Nanog (Fig. 4c green cluster, and Supplementary Data 1), suggesting that Phc1 represses this group of genes independently of Nanog. Taken together, our data demonstrate that Phc1 regulates the transcriptional network of mESCs through both Nanog-dependent and -independent repression and activation function. To determine whether Nanog and Phc1 form a transcriptional regulatory loop, we analyzed Nanog and Oct4 ChIP-seq data in mESCs and found that both Oct4 and Nanog bound to the promoter region of Phc1 (Supplementary Fig. 6a)[8]. However, knockdown of Pou5f1, but not Nanog, robustly reduced the transcript levels of Phc1 as shown by qPCR analysis (Supplementary Fig. 6b, c). These results indicate that both Phc1 and Nanog are direct transcriptional targets of Oct4 rather than forming a transcriptional regulatory loop.

**PHC1 interacts with NANOG**. Since our results demonstrated that Phc1 controls pluripotency of mESCs partly through Nanog, it prompted us to further investigate the relationship between Phc1 and Nanog. Previous Nanog interactome studies have shown that Nanog interacts with various TFs and chromatin-modifying complexes to regulate self-renewal of ESCs[38–40]. To test if Phc1 interacts with Nanog in ESCs, we performed co-immunoprecipitation (Co-IP) experiments and observed that endogenous PHC1 could precipitate endogenous NANOG, but not OCT4 and SOX2 in hESCs, whereas endogenous RING1B was not able to precipitate OCT4, SOX2, and NANOG except PHC1 (Fig. 5a). These results demonstrated that PHC1 interacts with NANOG to form a complex separate from cPRC1 in ESCs. We generated PHC1-FLAG and NANOG-HA plasmids, and Co-IP in HEK293T cells confirmed the interaction of exogenous PHC1 with exogenous NANOG (Fig. 5b). Moreover, endogenous Nanog could precipitate Phc1, but not Ring1b and Rybp, in mESCs transfected with a Phc1 overexpression plasmid (Fig. 5c).

**PHC1 organizes chromatin interactions of the *Nanog* locus**. At the 3D genome level, recent Hi-C data have shown that PHC1 promotes the formation of chromatin loops for PRC1-bound developmental genes in the genome[18–22]. Previous studies have shown that master TFs including Nanog bound to their own SEs as well as SEs of other pluripotency-associated genes thereby stabilizing promoter–enhancer looping and shaping 3D chromatin landscape for transcriptional regulation[7,8,41]. The interaction of Phc1 with Nanog independent of PRC1 in our study motivated us to look into whether Phc1 is involved in organizing the chromatin interactions of the *Nanog* locus in ESCs. We performed circularized chromosome conformation capture sequencing (4C-seq) of the genome-wide regions interacting with the *Nanog* locus in both $Phc1^{+/+}$ and $Phc1^{-/-}$ mESCs. By doing so, we showed that ablation of *Phc1* impaired the genome-wide chromatin interactions of the *Nanog* locus in mESCs (Supplementary Fig. 7b and Supplementary Data 2). Depletion of *Phc1* altered the intra-chromosomal interactions of *Nanog* promoter on chromosome 6 specific to mESCs (Supplementary Fig. 7a–d). In particular, previously known contact regions such as *Slc2a3* and 45 kb upstream SE of the *Nanog* locus exhibited significantly reduced interactions in $Phc1^{-/-}$ mESCs in comparison to $Phc1^{+/+}$ control (Fig. 5d). This result was consistent with the decreased contacts at these regions observed in the *Nanog* knockdown mESCs compared to the control after analysis of the previously published 4C-seq data[41] (Fig. 5d). The 45 kb upstream SE of the *Nanog* locus which is defined by Nanog binding and high H3K27ac signals as shown by ChIP-seq data analysis was previously reported to activate *Nanog* transcription through enhancer–promoter

looping[42]. This supports that Phc1 cooperates with Nanog to activate the transcription of *Nanog* by promoting its enhancer–promoter looping. Taken together, our results demonstrated that PHC1 interacts with NANOG to from a complex separate from cPRC1 which stabilizes genome-wide chromatin interactions of the *Nanog* locus to maintain pluripotency.

**Discussion**

This study uncovered the role of Phc1, a PcG protein, in maintaining pluripotency of ESCs. Our results demonstrated that Phc1 controls pluripotency not only through PRC1-dependent repression of developmental genes, but also via positive regulation of Nanog. Overexpression of Nanog can partly rescue loss of pluripotency induced by *Phc1* deficiency. Importantly, PHC1 interacts with NANOG to form a separate complex which activates *Nanog* transcription through stabilizing intra- and inter-chromosomal interactions of the *Nanog* locus (Fig. 6, proposed model).

Our results showed that Phc1 expression was highly enriched in both mouse and human PSC lines and epiblast in the early embryos, and knockdown or knockout of *Phc1* in hESCs or mESCs compromised pluripotency. This suggests that Phc1 plays an evolutionarily conserved role in maintaining pluripotency. In line with this, knockout of *Phc1* results in embryonic or perinatal lethality with neural developmental defect[43,44]. A point mutation in the SAM of human PHC1, a critical region for cPRC1-mediated gene silencing, has been implicated in causing primary microcephaly, a developmental disorder of neural progenitors[45]. Consistent with this, Phc1-bound genes with H2AK119ub1 enrichment in mESCs are involved in nervous system development and embryonic morphogenesis, and ablation of *Phc1* led to their derepression in our study, thus supporting that Phc1 maintains pluripotency through PRC1-dependent repression of developmental genes[16,17]. However, we also found that PHC1 immunostaining signals did not completely overlap with RING1B, and this finding was supported by Phc1-bound genes devoid of Ring1B occupancy and H2K119ub1 enrichment, thus indicating that Phc1 unlikely functions exclusively in a PRC1-dependent repression. Instead, a large percent of Phc1 targets which are involved in pluripotency are enriched with active H3K27ac and their expression decreased in *Phc1*-deficient mESCs, therefore pointing out that Phc1 also maintains pluripotency in a PRC1-independent manner by activating pluripotent genes. Our findings are supported by recent studies reporting gene activation function of PcGs during development and pluripotent cell fate decisions[23–31].

We found that either suppression or depletion of *Phc1* specifically decreases *Nanog* expression in hESCs and mESCs, respectively, which is indicating that Phc1 positively regulates *Nanog* transcription through direct and/or indirect mechanisms. *Nanog* locus is engaged in pluripotency-specific intra- and inter-chromosomal associations which is dependent on core TFs such as Nanog[41,46]. *Nanog* locus on murine chromosome 6 contains three SEs ($-45$ kb upstream, $-5$ kb upstream, and $+60$ kb downstream), and the $-45$ kb upstream and $-5$ kb upstream, but not $+60$ kb downstream SEs have been shown to regulate *Nanog* transcription through looping in ESCs[42]. We showed that PHC1 interacts with NANOG and promotes the genome-wide chromatin interactions of the *Nanog* locus (Fig. 5). Given the established role of PHC-SAM in forming chromatin loops[18–22], our data support that Phc1 stabilizes the chromatin topology of the *Nanog* locus for transcriptional activation. It remains unclear if Phc1 achieves this by binding to a consensus DNA sequence or non-coding RNA through its nucleic acid-binding FCS (Phe-Cys-Ser) domain for the target specificity[47].

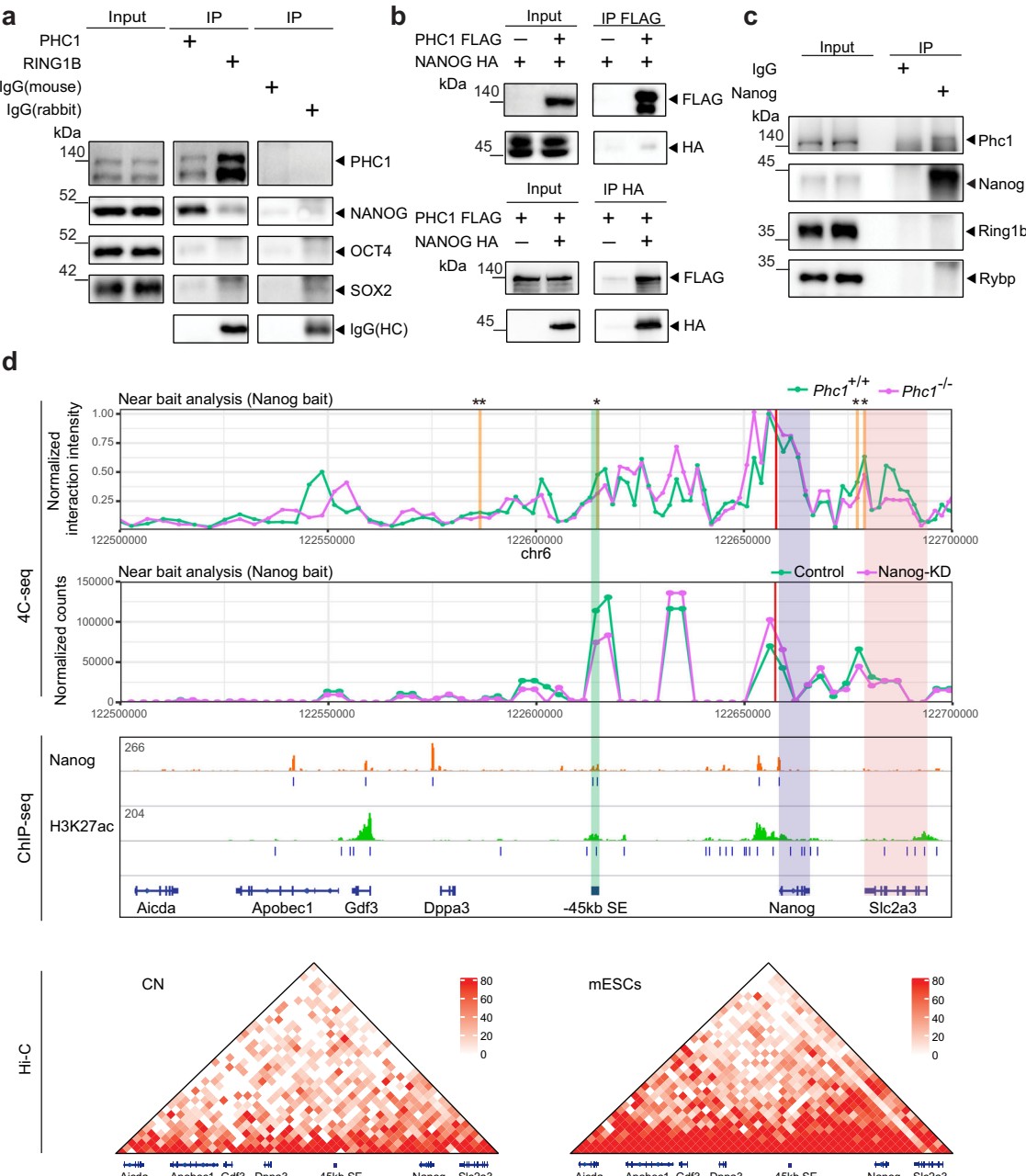

**Fig. 5 PHC1 interacts with NANOG and organizes genome-wide chromatin interactions of the *Nanog* locus. a** Co-immunoprecipitation (Co-IP) from total hESC extracts using antibodies against RING1B and PHC1 followed by immunoblotting of different proteins. **b** Co-IPs of nuclear extracts from HEK293T cells transfected with PHC1-FLAG and NANOG-HA followed by immunoblotting. **c** IP of nuclear extracts from mESCs using an antibody against Nanog and immunoblotting of Phc1, Ring1b, Rybp, and Nanog, respectively. **d** Top panel: Interaction profile showing the normalized interaction intensity (*y* axis) of regions with the *Nanog* promoter (anchor) in chr6: 122500000-122700000 of mm9 genome assembly (*x* axis) in $Phc1^{+/+}$ (green line) and $Phc1^{-/-}$ (purple line) mESCs. Regions with significant decreases in $Phc1^{-/-}$ than $Phc1^{+/+}$ mESCs were highlighted and statistical significance was indicated with asterisk. $n = 4$ independent experiments. Paired single-side *t*-tests were used (from left to right *p* values are 0.0090, 0.0234, 0.0379, and 0.0304, respectively). Middle panel: Analysis of the previously published 4C-seq datasets showing interaction profile of genomic regions near the *Nanog* promoter (empirical anchor) in chr6: 122500000-122700000 of mm9 genome assembly in control (green line) and Nanog KD (purple line) mESCs[41]; H3K27ac and Nanog ChIP-seq binding profiles in the corresponding regions[8,37]. Bottom panel: Hi-C interaction matrices of mouse cortical neurons (CNs) and mESCs in chr6: 122500000-122700000 of mm9 genome assembly, respectively[58]. Source data are provided as a Source Data file.

In summary, our study reveals the role of Phc1 in maintaining pluripotency partly through positively regulating *Nanog* transcription and proposes a mechanistic paradigm on transcriptional activation by Phc1. This significantly extends our knowledge on PcG-mediated regulation of gene expression. The concept of dual function of PcG proteins to regulate cell fate decision by cooperating with TFs has a broad application beyond pluripotency maintenance. It provides insights into understanding the functions of PcG proteins in cell fate decision, normal development, and human diseases.

## Methods

**Animal study**. Mice were bred in the Experimental Animal Facility of Zhejiang University. The experimental protocol and ethics were approved by the Animal

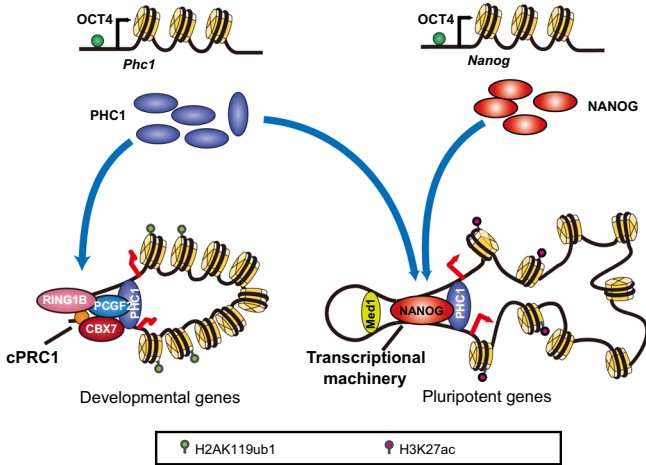

**Fig. 6 Proposed model.** *Phc1*, similar to *Nanog*, is transcriptionally activated by Oct4. While PHC1 exerts PRC1-dependent repression of development genes by associating with other cPRC1 subunits such as RING1B, CBX7, and PCGF2, it also interacts with Nanog to regulate chromatin landscape of the *Nanog* locus and activate pluripotent genes.

Care Facility of Zhejiang University (ZJU20200034). All in vivo experiments were performed in accordance with protocols from the Institutional Animal Care. Day 4.5 mouse embryos were collected from mice for immunofluorescent staining. H9 hESCs were injected into NOD/SCID mice for teratoma formation.

**Cell culture.** H9 hESCs (WiCell, WA09) and human iPSCs were cultured on Matrigel (BD Biosciences)-coated plates in complete mTeSR1 medium (STEM-CELL Technologies)[48,49]. HFF (ATCC, SCRC-1041), HEK293T (ATCC, CRL-11268), and MEF cells (Innovative Cellular Therapeutics Co, Ltd, 0304-500) were cultured in DMEM-high glucose (Corning), supplemented with 10% fetal bovine serum (FBS, Gibco). Mouse E14 ESCs (ATCC, CRL-1821) were maintained on 0.1% gelatin (Yisen)-coated plates in DMEM-high glucose, supplemented with 15% FBS, GlutaMAX (Gibco), non-essential amino acids (Gibco), penicillin/streptomycin (Gibco), 0.1 mM β-mercaptoethanol (Sigma), and 1000 U/mL leukemia inhibitory factor (LIF, Millipore).

**Gene knockout in mESCs and hESCs.** We used CRISPR/Cas9-based genome editing approach to knockout *Phc1* in mESCs and hESCs. To generate *Phc1*-deficient mESC lines, cells were transfected with px330 vector containing targeting sequences using Lipofectamine 2000 (gRNA sequences were listed in Supplementary Table 3). After selection with 2 μg/mL puromycin for 2 days, puromycin-resistant colonies were manually picked, expanded, and subjected to knockout validation by PCR and immunoblotting analysis. Multiple *Phc1*-deficient mESC lines were used for subsequent experiments. To knockout human *PHC1* gene in hESCs, gRNA sequences were cloned into the lentiCRISPR v2 vector (gRNA sequences were listed in Supplementary Table 3). Lentiviruses were produced by co-transfection of lentiCRISPR v2, psPAX2, and pMD2.G into 293T cells, and viral supernatants were collected at 24 and 48 h after transfection. hESCs were infected with the lentiviruses and selected with 2 μg/mL of puromycin. Bulk hESCs were subjected to PHC1 immunoblotting for validation.

**Stable gene knockdown and overexpression in mESCs.** Sequences of shRNAs against target genes were cloned into pLKO lentiviral vectors (shRNA target sequences were listed in Supplementary Table 3). Lentiviruses were produced by co-transfection of pLKO-shRNA, psPAX2, and pMD2.G into 293T cells, and viral supernatant were collected at 24 and 48 h after transfection. To overexpress target genes in E14 mESCs, gene sequences were cloned into pB-CAG vectors then transfected into cells by using Lipofectamine 2000 (Life Technologies). After knockdown or overexpression, the cells were cultured under 2 μg/mL of puromycin and 200 μg/mL of hygromycin, respectively.

**Human iPSCs generation.** Human reprogramming was performed by over-expression of four Yamanaka factors[49]. Briefly, HFFs were transduced in 6-well plates using OCT4, SOX2, KLF4, and c-MYC (OSKM) lentivirus supernatants produced in 293T cells. At 12 h after infection, cells were changed to DMEM-high glucose medium supplemented with 20% FBS, GlutaMAX, non-essential amino acids, penicillin/streptomycin, 0.1 mM β-mercaptoethanol, and 8 ng/mL basic fibroblast growth factor (bFGF, Life Technologies). At day 6 after infection, cells were then reseeded on feeder at the density of 30,000 cells per well of 6-well plate and switched to DMEM/F12 medium (Gibco) supplemented with 20% KnockOut

Serum Replacement (Gibco), GlutaMAX, non-essential amino acids, penicillin/streptomycin, 0.1 mM β-mercaptoethanol, and 8 ng/mL bFGF at D18. About 4 weeks after infection, ESC-like colonies were picked, characterized, and expanded for further experiments.

**Teratoma formation and histological analysis.** Control and PHC1 knockdown hESCs at 70–80% confluency were dissociated with EDTA and then resuspended with ice-cold 50% Matrigel/mTeSR medium. Then, $1 \times 10^6$ cells were subcutaneously injected into three groups of 4 male NOD/SCID mice. Injected mice were assessed regularly for the formation of tumors. At 4 weeks after injection, tumors were removed from the mice and their sizes from different groups were measured and quantified. Histological analysis was performed at the histology core facility at School of Medicine, Zhejiang University.

**Immunofluorescence.** For immunofluorescent staining, cells were washed twice in PBS and fixed in 4% paraformaldehyde for 15 min at room temperature. The cells were blocked with blocking buffer (1× PBS / 5% goat serum / 0.3% Triton X-100) for 60 min and then incubated with primary antibodies in dilution buffer (1× PBS / 1% BSA / 0.3% Triton X-100) at 4 °C overnight. The cells were subsequently stained with goat anti-rabbit or anti-mouse secondary antibody (Beyotime) at 1:300 for 60 min at room temperature in the dark. The primary antibodies NANOG (Cell Signaling Technology, 4903), OCT4 (Cell Signaling Technology, 2750), SOX2 (Cell Signaling Technology, 23064), RING1B (Cell Signaling Technology, 5694), RING1B (Abcam, ab181140), and PHC1 (Active motif, 39723) were all used at the concentration of 1:100. Immunofluorescent images were acquired by using Olympus BX61 confocal microscope and Nikon SIM microscope. ImageJ software was used to quantify the immunofluorescent intensities. To determine high and low signals, the average immunofluorescent intensities of sh*Scr* were set as the control. Intensities of PHC1 and NANOG staining in sh*PHC1*.1 and sh*PHC1*.2 groups higher or lower than the corresponding controls were defined as PHC1^highNANOG^low and PHC1^lowNANOG^low, respectively. The percentage of PHC1^highNANOG^low and PHC1^lowNANOG^low cells were quantified relative to the sh*Scr* control. For the quantification of PHC1 and RING1B co-immunostaining, super-resolution microscopic images were analyzed with Imaris software. PHC1+RING1B+ and PHC1+RING1B− foci that represent PHC1 signals overlapping with and without RING1B, respectively, in more than 40 cells from the staining experiments with two different RING1B antibodies were quantified.

**AP staining.** In all, $1 \times 10^3$ single E14 mESCs of different conditions were seeded in a 6-well plate and 1 or $2 \times 10^3$ hESCs were plated into a 6 cm dish. Ten days after seeding mESCs and 14 or 18 days for hESCs, alkaline phosphatase staining was performed with an alkaline phosphatase detection kit (Beyotime) according to the manufacturer's instructions. The number of colonies was then quantified after the staining.

**Flow cytometry.** mESC line carrying the *Nanog*-GFP reporter (a gift from Jin Zhang Lab at Zhejiang University School of Medicine) was infected with sh*PHC1* or control lentiviruses. The cells were selected in the culture medium supplemented with 2 μg/mL puromycin for 2 days. Then the cells were harvested by incubation in 0.05% trypsin-EDTA (Gibco) and washed once with PBS. Flow cytometry data were collected on a BD Cytomics FC 500MCL machine and analyzed with FlowJo software (v7.6) or CXP software (v2.3).

**Chromatin immunoprecipitation.** In all, $0.5–1 \times 10^7$ cells were fixed in 1% formaldehyde for 10 min at 37 °C and then the unreacted formaldehyde was quenched by adding glycine to a final concentration of 125 mM. Cells were lysed in the SDS lysis buffer (1% SDS, 10 mM EDTA, 50 mM Tris, pH 8.1) with protease inhibitors (Sigma). The cell lysate was sonicated to shear cross-linked DNA to 200–1000 bp in length by using a BioRupter sonicator (Diagenode). Chromatin was diluted with 10× ChIP Dilution Buffer (0.01% SDS, 1.1% Triton X-100, 1.2 mM EDTA, 167 mM NaCl, 16.7 mM Tris-HCl pH 8.1) containing protease inhibitors. The chromatin lysate was incubated with the immunoprecipitating antibodies overnight at 4 °C with rotation. Protein–antibody complexes were bound by incubating with Protein A/G Dynabeads (Invitrogen) for 3–4 h at 4 °C. Samples were then washed with 1 mL of each of the buffers listed in the given order: low-salt washing buffer (1% Triton X-100, 2 mM EDTA, 20 mM Tris-HCl pH 8.1, 150 mM NaCl), high-salt washing buffer (0.1% SDS, 1% Triton X-100, 2 mM EDTA, 20 mM Tris-HCl pH 8.1, 500 mM NaCl), LiCl buffer (0.25 M LiCl, 1% IGEPAL-CA630, 1% deoxycholic acid, 1 mM EDTA, 10 mM Tris pH 8.1), and TE buffer (10 mM Tris-HCl, 1 mM EDTA, pH 8.0). The beads were resuspended in TE buffer supplemented with 0.25% SDS and proteinase K. After incubation overnight at 65 °C, a Qiagen PCR Purification Kit was used to purify the DNA. ChIP-qPCR primers were listed in the Supplementary Table 2. The following antibodies were used for ChIP experiments: Control mouse IgG (Sigma, I8765), control rabbit IgG (Cell Signaling Technology, 2729), H2AK119ub1 (Cell Signaling Technology, 8240, 1:100), Nanog (Cell Signaling Technology, 8822, 1:100), and PHC1 (Cell Signaling Technology, 13768, 1:50).

**Protein co-immunoprecipitation and western blotting**. For Co-IP experiments, 293T cells ($2 \times 10^6$) were plated in a 10 cm plate. Next day, the 293T cells were transfected with HA-NANOG and FLAG-PHC1 plasmids and polyethylenimine (PEI, 2 mg/mL) transfectant mixture. At 48 h after transfection, cells were collected and nuclear lysates were extracted by a Nuclear Protein Extraction KIT (Beyotime). The lysates were then incubated with anti-FLAG Affinity Gel (Bimake, B23101) or anti-HA magnetic beads (Bimake, B26202) overnight at 4 °C. Beads were washed six times with wash buffer (150 mM NaCl, 50 mM Tris, 1 mM MgCl₂, 1% NP-40, PH7.5). Proteins were eluted, denatured at 100 °C for 10 min, and analyzed by SDS-PAGE. For IP of endogenous proteins in hESCs, cells were collected and lysed with lysis buffer. Cell lysates were first incubated with antibodies against target proteins overnight at 4 °C. Next day, the mixture was incubated with Protein A/G magnetic beads (Life Technologies) and rest of the steps were following the same procedure as Co-IP experiment. Western blotting was performed according to standard procedures. For IP in mESCs, the cells were overexpressed with exogenous Phc1-FLAG, collected, and lysed with lysed buffer. The cell lysates were then washed with PBS three times, fixed in PBS with 0.25 μg/mL DTBP for 40 min at room temperature to cross-link the proteins. The lysates were then centrifuged and a Nuclear Protein Extraction KIT (Beyotime) was used to extract nuclear proteins, and the nuclear lysates were then incubated with the anti-Nanog antibody at 4 °C overnight. Next day, the mixture was incubated with Protein A/G magnetic beads (Life Technologies) and beads were washed six times in lysis buffer as above. Immunoprecipitated proteins were suspended in 1× LB with 15 mM DTT at 37 °C for 40 min to reverse cross-linked proteins, and the proteins were denatured at 100 °C and western blotting was performed according to standard procedures.

**Plasmids**. The coding sequence of *Nanog* gene was amplified from cDNA obtained from mESCs and cloned into the PiggyBac transposon pPB-CAG-3×FLAG-pgk-hph vector, a gift from Jin Zhang (Zhejiang University School of Medicine, China). HA-NANOG and FLAG-PHC1 sequences were cloned into pcDNA5 vector from Addgene for Co-IP experiments. shRNA inserts were cloned into pLKO lentiviral vector (Addgene). The gRNA sequences targeting mouse *Phc1* gene in mESCs were cloned into px330-U6-Chimeric_BB-CBh-hSpCas9-puro vector from Yan Zhang (Institute Pasteur of Shanghai, Chinese Academy of Science, China). The gRNA sequence targeting human *PHC1* gene in hESCs was cloned into lentiCRISPR v2 vector (Addgene). shRNA target sequences were listed in Supplementary Table 3.

**RNA isolation, RT-PCR, and RNA-seq**. Total RNA was extracted with TRIzoI (Takara) following standard procedure. cDNA was reverse transcribed using ReverTra Ace® qPCR RT Master Mix (TOYOBO). RT-qPCR reactions were performed using SYBR® Premix Ex TaqTM kit (Takara) and the LightCycler480 machine (Roche). Samples were run in triplicates and expression was normalized to the housekeeping gene *Actin*. All the primers used were listed in Supplementary Table 1. RNA-seq library preparation and sequencing were performed by BGI Group in Shenzhen, China.

**RNA-seq and ChIP-seq analyses**. RNA-seq reads were aligned to the mm10 mouse genome assembly using Bowtie2 (v2.2.5)[50]. RSEM (v1.2.15) was run to quantify the expression FPKMs of each annotated transcript RefSeq. A gene set with a 1.2-fold expression difference and adjusted $p$ value $\leq 0.001$ were considered as being differentially expressed. The expression levels of all genes were shown as heatmaps. Differential genes that were up- or down-regulated in $Phc1^{-/-}$ mESCs compared with the $Phc1^{+/+}$ mESCs were clustered using K-means clustering algorithm across all samples.

Raw data from the previously published ChIP-seq datasets GSE72886 (for H3K27ac), GSE44288 (for Nanog), and GSE89949 (for H2AK119ub1, Phc1, and Ring1b) were downloaded from EMBL-EBI (Supplementary Table 7). All these data were re-analyzed according to the same criteria. Single-end ChIP-seq data were aligned to the mm9 mouse genome assembly using Bowtie2 (v2.3.4.2) by default[50]. Peak calling was performed by MACS2 program (v2.1.1.20160309) using corresponding input samples for background normalization[51]. The option of cut-off value analysis in MACS2 was used to decide an appropriate cut-off (for Phc1 $p = 1.00e-02$, for Ring1b $q = 1.00e-08$, and for Nanog, H3K27ac and H2AK119ub1 $q = 1.00e-03$). Each peak was annotated with its nearest gene using the R (v3.4.0) package ChIPseeker (v1.10.3)[52]. Genomic annotation and gene coordinates were obtained from a BioConductor package TxDb.Mmusculus.UCSC.mm9.knownGene. The resulting normalized signal enrichment file in bigWig format was visualized on the Integrative Genomics Viewer (IGV) (v2.4.14)[53]. The other published ChIP-seq data in the Gene Expression Omnibus (GEO) under the accession numbers GSE44288 (for OCT4) and GSE104690 (for H2AK119ub1 and RING1B of hESCs) in bigWig format were visualized directly in IGV.

**4C-seq**. $Phc^{+/+}$ and $Phc1^{-/-}$ mESCs were prepared for 4C as previously described[54]. Briefly, $2 \times 10^6$ cells were cross-linked with 1% formaldehyde. The cross-linked pellet was then resuspended with lysis buffer (10 mM Tris-HCl pH 8, 10 mM NaCl, 0.2% Igepal CA630, Protease Inhibitor). The suspension was centrifuged and the supernatant was discarded. The chromatin was then digested with 200 units DpnII after pre-treatment with 0.5% SDS and 1.14% triton X-100. The reaction was heat-inactivated and then incubated with ligation buffer (1× T4 Ligase

buffer, 8.3% TritonX-100, 120ug BSA, 10000U T4 DNA Ligase) at 16 °C overnight. Chromatin was reverse cross-linked at 68 °C for 60 min with additional NaCl after Proteinase K and SDS treatment. The DNA was then purified by DNA clean beads followed by the second digestion with NlaIII and ligation in 10 mL 1× ligation buffer. DNA was precipitated with acetate and ethanol. The precipitate was purified again by DNA clean beads. Amplification of 4C products by PCR and paired-end libraries was performed and then sequenced on the Illumina Novaseq platform. Primers for 4C were listed in Supplementary Table 5.

4C-seq was performed on four biological replicates.

**Analysis of 4C-seq data**. Here, 4C data were processed using 4C-ker package (v1.0) according to the software's instructions[55]. Briefly, the reduced genome (mm9) consisting of only 25 bp sequences flanking the primary restriction enzyme sites was generated using a customized script, and 4C-seq reads were trimmed and mapped to this reduced genome by Bowtie2 (v2.3.4.1)[50]. Each sample's count profiles were created from mapped data and removed self-ligated and undigested fragments around bait fragment. Then the count profiles were used to analyze near *cis*-, far *cis*-, and *trans*-interactions by 4C-ker. The far *cis*- and *trans*- interaction profiles were visualized by ggbio (v1.30.0)[56]. The Sequence Read Archive (SRA) files of 4C-seq data of control (BioSample ID: SAMN02222862) and Nanog knockdown mESCs (BioSample ID: SAMN02222865) were downloaded from the National Center for Biotechnology Information Gene Expression Omnibus (GEO) under accession number GSE37275. These 4C-seq data were analyzed as described above. For intra-chromosomal interactions, 4C-seq data were analyzed on genomic regions near the *Nanog* promoter (anchor) in chr6: 122500000-122700000 of mm9 genome assembly containing the *Nanog* −45 kb SE (chr6: 122612514-122614260, mm9) as previously defined[8].

**Hi-C data analysis**. The SRA files of high-resolution mESCs (BioSample ID: SAMN06564305) and mouse cortical neurons (CNs) (BioSample ID: SAMN06564287) Hi-C data were downloaded from the National Center for Biotechnology Information Gene Expression Omnibus (GEO) under accession number GSE96107. After the SRA files were gathered, the archives were extracted and saved in FASTQ format using the SRA Toolkit (v2.9.0). The paired-end reads of fastq files were aligned, processed, and iteratively corrected using HiC-Pro (v 2.11.1) as previously described[57]. Briefly, short sequencing reads were first independently mapped to mouse mm9 reference genome using the bowtie2 aligner with end-to-end algorithm and '-very-sensitive' option. To rescue the chimeric fragments spanning the ligation junction, the ligation site was detected and the 5′ fraction of the reads was aligned back to the reference genome. Unmapped reads, multiple mapped reads, and singletons were then discarded. Pairs of aligned reads were then assigned to DpnII restriction fragments. Read pairs from the uncut DNA, self-circle ligation, and PCR artifacts were filtered out and the valid read pairs involving two different restriction fragments were used to build the contact matrix. Valid read pairs were then binned at a 5 kb resolution by dividing the genome into bins of equal size. To eliminate the possible effects of variable sequencing depths on data analyses, we randomly sampled equal numbers of read pairs from each condition for downstream analyses involving comparison between conditions. The binned interaction matrices were then normalized using Knight-Ruiz matrix balancing method. Visualization of normalized Hi-C matrix and topologically associated domains was carried out by HiCExplorer (https://hicexplorer.readthedocs.io/en/latest/).

**Statistics and reproducibility**. Data of bar charts are represented as mean ± s.e.m. or mean ± s.d. For violin plot, the central dotted line represents the median. For the quantification results, *n* values refer to independent experimental replicates or sample sizes in each figure legend. Significance was tested using two-tailed unpaired Student's *t*-tests or one-way ANOVA test with Bonferroni's multiple comparison. Both the Pearson correlation coefficient and Spearman correlation coefficient were computed to assess the correlation of log expression values of *NANOG* with *PHC1* and *RING1B* in each cell in Supplementary Fig.1b. The correlation was performed using linear regression, and 95% confidence interval was denoted. Figure 1a is representative of three (PRC2 genes) or four (PRC1 genes) independent experiments. Figure 1b is representative of more than three independent experiments. Figure 1c is representative of three independent experiments. Figure 1e is representative of two independent staining (about 20 embryos were stained in total). Figure 2a, b, f, g is representative of three independent experiments. Figure 3a, b, d, e is representative of three independent experiments. Figure 3c, g is representative of four independent experiments. RNA-seq data in Fig. 4c were from experiments performed in three biological replicates, and 4C-seq data in Fig. 5d were from experiments performed in four independent replicates. Figure 5a is representative of two independent experiments. Figure 5b, c is representative of three independent experiments. Supplementary Fig. 2a, b is representative of three independent experiments. Supplementary Fig. 2c is representative of four independent experiments. Supplementary Fig. 3a, e is representative of three independent experiments. Supplementary Fig. 3g is representative of four independent experiments. Supplementary Fig. 3f is representative of two independent experiments. Supplementary Fig. 5a, b is representative of results of one experiment. Supplementary Fig. 5c is representative of four independent experiments. Supplementary Fig. 5d, e is representative of three independent experiments. Supplementary Fig. 6b, c is representative of three

independent experiments. Supplementary Fig. 7a, b is representative of four independent experiments.

**Reporting summary**. Further information on research design is available in the Nature Research Reporting Summary linked to this article.

## Data availability

RNA-seq data are available in the SRA database with the accession number PRJNA532733 and all the 4C-seq data were deposited with GEO accession GSE155524. Previously published ChIP-seq data that were re-analyzed in this study are available according to the GEO accession numbers that were listed in Supplementary Table 7. Previously published Hi-C data that were re-analyzed here are available under the accession code GSE96107. All other data supporting the findings of this study are available on reasonable request from corresponding author. Source data are provided with this paper.

## Code availability

All codes used in this study for bioinformatics analysis are available upon reasonable request from corresponding author.

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

## Acknowledgements

We thank Dr. Xiaokang Zhang in the Center of Cryo-Electron Microscopy (CCEM), Zhejiang University for his technical assistance on computer clustering. We are also grateful to Dr. Yan Zhang at Institute Pasteur of Shanghai, Chinese Academy of Science for providing plasmid and technique to support gene targeting in mESCs. We are deeply thankful for the computational resources provided by the Super Computer System, Human Genome Center, Institute of Medical Science, University of Tokyo, Japan. This work was supported by grants from the National Key Basic Research Program of the Ministry of Science and Technology of China (2015CB964901), the National Natural Science Foundation of China (91849131 and 31271594), Zhejiang Provincial Natural Science Foundation of China (LZ21C120001 and LY17C060003), the International S&T Cooperation Program of the Ministry of Science and Technology of China (2014DFG32790), Opening Foundation of Key Laboratory of the diagnosis and treatment research of reproductive disorders of Zhejiang Province, Shenzhen Public Service Platform of Molecular Medicine in Pediatric Hematology and Oncology, Shenzhen Fund for Guangdong Provincial High Level Clinical Key Specialties (SZGSP012), Shenzhen Key Medical Discipline Construction Fund (SZXK034), the National Key R&D Program of China, Stem Cell and Translation Research (2018YFA0109300), Zhejiang Province Science Foundation for Distinguished Young Scholars (LR19H080001), the National Natural Science Foundation of China (81870080, 31671537) and the Innovative Team Program from the Bioland Laboratory (Guangzhou Regenerative Medicine and Health Guangdong Laboratory) (2018GZR110103001).

## Author contributions

J.J., F.W., and C.Li conceived and developed the original idea. L.C., Q.T., X.C., Q.Z., L.S., C.Liu, B.G., Y.Z., X.J., and W.L. performed the experiments. L.F. and G.G. contributed to the analysis of single-cell sequencing. C.Li, L.C., P. J., H. Y., J. Z., P. Q., and C.Liu contributed to the bioinformatics analysis. P.J. and H.Y. analyzed 4C-seq and Hi-C data, respectively. J.J., F.W., and C. Li designed and supervised the study. J.J. and F.W. provided funding support on this work. J.J. wrote the manuscript, and L.C., Q.T., P.J., H.Y., L.F., G.V., M.A., M.A.E., Q.S., and D.N. contributed to the writing of the manuscript.

## Competing interests

The authors declare no competing interests.
