## [Peer Review File · Nature Communications]

Reviewers' comments:

Reviewer #1 (Remarks to the Author):

The main conclusion of this paper is that PHC1 contributes to the pluripotent state of ES cells (mouse and human) by activating expression of Nanog through long range organization of chromatin. This activation function is independent of PRC1.

Overall, the conclusion that Nanog expression is decreased when PHC1 expression is decreased is supported by the data.

I have several major concerns with the evidence for the other key conclusions.

1) The conclusion that PHC1 activates Nanog expression independent of PRC1 requires additional biochemical and genetic evidence. The conclusion is currently based on the super-resolution microscopy shown in Supplementary Fig. 2a, and analysis of overlapping peaks from publicly available ChIP-seq data.

A) The authors state "Super-resolution microscopic imaging analysis of PHC1 and RING1B co-immunostaining showed that, while the majority of PHC1 and RING1B signals overlapped, some did not co-localize with each other (Supplementary Fig. 2a). »

To conclude that PHC1 does not co-localize with RING1B from microscopy analysis, this analysis would need to be quantitative, and also include controls demonstrating that failure to colocalize cannot be explained by the efficiency of antibody labeling, binding, or detection. Currently, there is a single image in the supplemental data (Fig. S2A). The arrowheads in the merged image that should show non-colocalizing signals are not clearly positive for either PHC1 or RING1B. These data provide no evidence for the conclusion.

Ideally this analysis should be backed up with biochemistry—for example a glycerol gradient or size exclusion column of extracts from these cells showing that a fraction of PHC1 migrates in a distinct location from other PRC1 subunits.

B) For the bioinformatic argument, in supplementary Figure 2b, the authors should show example traces of non-overlapping PHC1 and Ring1B peaks, to ensure that the issue is not simply a sensitivity or peak calling issue. This is also a question in Fig. 4c-- the PHC1 signals for the representative traces from the blue and orange clusters are very low and seem to span the entire region shown—are the authors certain these are enriched for PHC1? Showing the peak calls may be helpful here, along with a PHC1 negative gene with the the same scale on the Y-axis. In fact, these data seem to support PHC1 possibly working with Nanog to repress gene expression (probably in the context of PRC1) since the traces shown representing the purple cluster (repressed by PHC1 and Nanog) are clear and convincing.

C) As part of the argument that PHC1 functions independently of PRC1, the authors analyze a small number of genes by Chip-qPCR for H2AK119ub1, and from this conclude that PHC1 knockdown does not affect PRC1 function (Fig.S2F). Are these genes normally bound by PHC1? It could be helpful to show browser traces of PHC1 and RING1B binding, along with H2Aub in wild type cells, and indicate where primer sequences are on these traces. Fig. S2F should also include negative controls—IgG and sites that are negative for H2AK119ub1.

D) A crucial test of the conclusion that PHC1 activity is independent of PRC1 is that PHC1-mediated activation of Nanog is expected to persist in the absence of other PRC1 subunits. This should be tested, by depleting Ring1B, confirming that PRC1 targets are derepressed, and testing whether Nanog levels are affected.

2) The authors state that "We found that Phc1 interacts with Nanog and activates Nanog transcription by stabilizing the genome-wide chromatin interactions of the Nanog locus. »

The causal link between PHC1-dependent chromatin organization and gene activation has not been

demonstrated—the events are correlated. PHC1 mediates activation through long range interactions activation is mediated through effects on long range organization is not warranted from the data, which are that Nanog expression is correlated with that of PHC1, and disrupting PHC1 disrupts long range interactions. A more in depth analysis of the long range interactions may partially address this—are there enhancers that normally interact with Nanog that no longer interact when PHC1 is knocked out? How does PHC1 binding relate to the long range contacts observed? Do long range contacts correspond to PHC1-only sites, or are they PRC1 sites?

3) The authors demonstrate that Nanog can be co-immunoprecipitated with PHC1, and claim that this interaction is independent of PRC1. The co-IP experiments shown in Fig. 5 provide evidence that Nanog can copurify with PHC1. However, the evidence that it does not co-purify with PRC1 (e.g. RING1B) is not strong given that there is a faint band in the RING1B IP in 5a, and this experiment was not done in 5c. The RING1B co-IP should also be done in the mESCs, especially since a cross-linking step was used in this case. These blots should be quantified and evidence for reproducibility provided.

4) In their model (Fig. 6), the authors imply that interactions between Nanog and PHC1 recruit PHC1 to change chromatin conformation to activate gene expression. However, no evidence is provided that the interaction between Nanog and PHC1 is involved in regulating transcription. Do Nanog and PHC1 co-localize at genes that are activated when PHC1 is knocked out? A critical test of this model would be to determine whether removing Nanog disrupts targeting of PHC1 to these genes, or conversely whether removing PHC1 disrupts Nanog targeting.

Additional points :

1) In Figure 1D, is it possible to plot the correlation between the two data sets? What is the basis for concluding that PHC1 and Nanog have similar expression patterns—they do overlap in some cases, but in others (especially later stages, where PHC1 is high but Nanog is low), they do not. This analysis, and the one in 1e would be more compelling if additional Polycomb proteins/genes were analyzed, as in 1a. (i.e. is the relationship between PHC1 and Nanog levels unique?).

2) The authors state that reducing PHC1 using shRNA does not affect PRC1 levels or activity, based on Fig. 1d. These blots need to be quantified, and evidence for reproducibility shown. It appears that the H2AK119ub1 signal is decreased in the shPHC1.2 cells. Please state how was Nanog quantified—was it normalized to the loading control?

3) How was immunofluorescence quantified and divided into high-low for the graph shown in Fig. 2G? This needs to be detailed in the methods.

4) In Figure 3, analysis of multiple clones should be shown, to confirm that the effects are not due to off-target editing. From the data in Fig. S3, it seems that the correlation between PHC1 knockdown and Nanog levels is not consistent. These data should be quantified. It also needs to be clarified if single clones were used for the experiments, or multiple clones, and if the data were pooled or handled separately.

5) Please explain how the colonies were scored for Fig. 3f.

6) The Nanog rescue experiments (ie Fig. 3e, Fig. 5) should also include a control for overexpression of Nanog in a wild-type background. The level of overexpression of Nanog should be quantified.

7) The authors conclude that Oct4, but not Nanog, regulate PHC1 expression. In the Western blot shown in Fig.S4b, it appears that PHC1 levels are increased in the shNanog cells—these results need to be quantified, and evidence for reproducibility shown.

8) From the methods section, it appears that the 4C-seq experiment was conducted a single time. This experiment needs to be replicated.

9) The methods section of this paper requires additional details. While the different steps involved in many of the protocols are well-described, critical details such as the amounts of reagents used and detailed protocols are not presented (for example, how cells were prepared for immunofluorescence and FACS analysis, how immunofluorescence experiments were quantified, how many cells were used for Co-IP experiments). Additional details on how the super-resolution microscopy data were collected

and analyzed should be provided. The bioinformatics analysis is also not described in sufficient detail—since analysis of publicly available data is critical to the authors argument, exactly how these data were handled and compared should be described (i.e. were peaks called to create Venn diagrams? how were peak calls done? were the data re-analyzed with the same pipeline to make comparisons?).

Reviewer #2 (Remarks to the Author):

In this manuscript, the authors described that Phc1, a subunit of Polycomb Repressive complex 1 (PRC1), is implicated in Nanog transcriptional activation, which in turn affect pluripotency. The mechanism reported here suggest that Phc1 interacts directly with Nanog to regulate Nanog's promoter chromatin architecture via stabilization of intra- chromosomal and inter-chromosomal interactions.

Overall this is an interesting manuscript, which unveils a novel PRC1-independent function for Phc1. Nevertheless, several experiments are necessary to fully support the authors' conclusions, as detailed below.

Main points:

- 1) To have a complete view of the expression changes of Polycomb subunits during differentiation, the authors should include in figure 1A and 1B the expression levels of PRC2 core components as well as of its associated factors.
- 2) Please define the y-axis for Figure 1D. Are the expression levels of Nanog and of PHC1 in the same range?
- 3) The resolution of the figure 2A is too low to appreciate any differences in morphology between WT and KD cells. This should be fixed.
- 4) At the end of page 5 the authors conclude that "suppression of PHC1 expression did not obviously change the levels of RING1B, the E3 ligase subunit of PRC1, and its associated histone modification H2AK119ub1". It seems to me that H2AK119ub1 levels are indeed lower in shPHC1.2. Please comments on this, or provide a more representative western blot panel.
- 5) It is unclear to me why the level of Nanog are much more effected upon CRISPR KO approach (Figure 2E) when compared to shRNA (Figure 2D), since in both cases the levels of PHC1 downregulation are similar.
- 6) The RNA-seq analysis provides important information. Yet this should be integrated with a ChIP-seq data obtained in the same samples. This is even more relevant since the authors relay for ChIP-seq data on a publication which is 15-years old (Boyer et al, 2005).
- 7) It is important that the interaction between Nanog and Phc1 does not involve other subunits of the PRC1 complex. For this an IP for Nanog should be performed followed by Western blot for PRC1 subunits.

Reviewers' comments:

Reviewer #1 (Remarks to the Author):

The main conclusion of this paper is that PHC1 contributes to the pluripotent state of ES cells (mouse and human) by activating expression of Nanog through long range organization of chromatin. This activation function is independent of PRC1.

Overall, the conclusion that Nanog expression is decreased when PHC1 expression is decreased is supported by the data.

I have several major concerns with the evidence for the other key conclusions.

1) The conclusion that PHC1 activates Nanog expression independent of PRC1 requires additional biochemical and genetic evidence. The conclusion is currently based on the super-resolution microscopy shown in Supplementary Fig. 2a, and analysis of overlapping peaks from publicly available ChIP-seq data.

A) The authors state “Super-resolution microscopic imaging analysis of PHC1 and RING1B co-immunostaining showed that, while the majority of PHC1 and RING1B signals overlapped, some did not co-localize with each other (Supplementary Fig. 2a).” To conclude that PHC1 does not co-localize with RING1B from microscopy analysis, this analysis would need to be quantitative, and also include controls demonstrating that failure to colocalize cannot be explained by the efficiency of antibody labeling, binding, or detection. Currently, there is a single image in the supplemental data (Fig. S2A). The arrowheads in the merged image that should show non-colocalizing signals are not clearly positive for either PHC1 or RING1B. These data provide no evidence for the conclusion.

Ideally this analysis should be backed up with biochemistry—for example a glycerol gradient or size exclusion column of extracts from these cells showing that a fraction of PHC1 migrates in a distinct location from other PRC1 subunits.

Response: We thank the reviewer for the excellent comments and suggestions. To address the concern on antibody labeling efficiency, we have now performed co-immunostaining of PHC1 with an additional RING1B antibody in hESCs and quantified the staining. Our results below consistently showed that although majority (around 80%) of PHC1 signals overlapped with that of RING1B (PHC1⁺RING1B⁺), approximately 20% PHC1 signals did not appear to exhibit detectable RING1B signal (PHC1⁺RING1B⁻), regardless of different RING1B antibodies used. Thus, these results suggest that PHC1 associates largely, but not exclusively, with RING1B to exert its function in hESCs. It suggests that PHC1 possesses PRC1-independent function in the maintenance of pluripotency. These results have been updated in Supplementary Fig. 3b of the revised manuscript.

Moreover, we performed glycerol gradient (10-15%) biochemical experiments as this reviewer suggested. Our results showed that Phc1 migrated together with Nanog, but not with Ring1b, in fraction 6 of nuclear extracts from mESCs (highlighted by the red dotted line in panel **a** of the data below). Similarly, PHC1 migrated without detectable RING1B in fractions 6-8 of nuclear extracts from hESCs (shown by the red dotted line in panel **b** of the data below). Taken together, these results support that PHC1 possess both PRC1-dependent and -independent dual functions in pluripotent stem cells.

B) For the bioinformatic argument, in supplementary Figure 2b, the authors should show example traces of non-overlapping PHC1 and Ring1B peaks, to ensure that the issue is not simply a sensitivity or peak calling issue. This is also a question in Fig. 4c-- the PHC1 signals for the representative traces from the blue and orange clusters are very low and seem to span the entire region shown—are the authors certain these are enriched for PHC1? Showing the peak calls may be helpful here, along with a PHC1 negative gene with the same scale on the Y-axis. In fact, these data seem to support PHC1 possibly working with Nanog to repress gene expression (probably in the context of PRC1) since the traces shown representing the purple cluster (repressed by PHC1 and Nanog) are clear and convincing.

Response: We thank the reviewer for the comments and suggestions. We have now performed peak calling by MACS2 program using corresponding input samples for background normalization. We have now further analyzed the Phc1 and Ring1b ChIP-Seq data in mESCs and shown examples on non-overlapping Phc1 and Ring1b binding peaks of *Ano2* and *Foxj2* genes, respectively (see the data below). These results have been included as Supplementary Fig. 3c-d of the revised manuscript. Description of these results in the results section and legend of this Figure have also been revised accordingly in the manuscript.

To address the reviewer's comments on Fig. 4c, we have now re-analyzed ChIP-seq raw data using peak calling. Reanalysis results showed that there was a clear Phc1 and Nanog binding peak on *Ybx3* gene with active H3K27ac but without repressive H2AK119ub1 enrichment in the representative trace of orange cluster. Therefore, this orange cluster represents genes which are activated by Phc1 and Nanog in a PRC1-independent manner. Our results indicated that there was no enrichment of Phc1 and H2AK119ub1 binding to the blue cluster for example *Esrrb* which is enriched with Nanog and H3K27ac binding. Our RNA-seq data demonstrated that *Phc1* depletion reduced their expression which could be rescued by overexpression of Nanog, suggesting that Phc1 positively regulates their expression through Nanog. The purple cluster genes represent repression by Phc1 possibly in concert with Nanog in PRC1-dependent manner. These data precluded analysis sensitivity concerns and have now been updated in Figure 4 of the revised manuscript. The analysis details have also been provided in RNA-seq and ChIP-seq analysis in the methods section of the revised manuscript.

C

C) As part of the argument that PHC1 functions independently of PRC1, the authors analyze a small number of genes by Chip-qPCR for H2AK119ub1, and from this conclude that PHC1 knockdown does not affect PRC1 function (Fig.S2F). Are these genes normally bound by PHC1? It could be helpful to show browser traces of PHC1 and RING1B binding, along with H2Aub in wild type cells, and indicate where

primer sequences are on these traces. Fig. S2F should also include negative controls—IgG and sites that are negative for H2AK119ub1.

Response: We thank the reviewer for the comments and suggestions. We examined the binding of PHC1 to these genes by ChIP-PCR and our results showed that these genes are bound by PHC1 in hESCs (see the data below). While PHC1 ChIP-seq data in hESCs are not available, traces of RING1B binding and H2AK119ub1 on these genes in hESCs were shown in IGV images below and primers for ChIP-PCR analysis were also indicated. These results were included as Supplementary Fig. 4 in the revised manuscript.

Fig. S4: Binding of PHC1, RING1B and H2AK119ub1 to developmental genes in hESCs. ChIP-PCR analysis of PHC1 binding at target genes including *EOMES*, *GSX2*, *EN2*, *GATA4*, *GATA6*, and *GAPDH*, a non-target control, in wild type hESCs. IgG was used as the negative

control. IGV images of RING1B and H2AK119ub1 density at these genes were also shown. Primers used for H2AK119ub1 ChIP-PCR analysis on these genes in Fig.S3g were indicated.

Negative controls including shScr + IgG and promoter region of *GAPDH* negative for H2AK119ub1 have now been included in the H2AK119ub1 ChIP-PCR as the Supplementary Fig. 3g in the revised manuscript (see the data below).

D) A crucial test of the conclusion that PHC1 activity is independent of PRC1 is that PHC1-mediated activation of *Nanog* is expected to persist in the absence of other PRC1 subunits. This should be tested, by depleting *Ring1B*, confirming that PRC1 targets are derepressed, and testing whether *Nanog* levels are affected.

Response: We thank the reviewer for the comments and suggestions. As the reviewer suggested, we have now knocked down *Ring1b* in mESCs by three independent shRNAs and examined the expression levels of *Nanog* and PRC1 target genes including *Eomes*, *Gsx2*, *En2*, *Gata4* and *Gata6*. Our results below show that knocking down *Ring1b* induced derepression of a few PRC1 target genes such as *En2*, *Gata4* and *Gata6*, however, it did not significantly affect the expression of *Nanog* at both transcript and protein levels. These results support that PHC1 activates the expression of *Nanog* independent of PRC1. These results have been included as Supplementary Fig. 5c-d of the revised manuscript.

2) The authors state that “We found that Phc1 interacts with *Nanog* and activates *Nanog* transcription by stabilizing the genome-wide chromatin interactions of the *Nanog* locus.”

The causal link between PHC1-dependent chromatin organization and gene activation has not been demonstrated—the events are correlated. PHC1 mediates activation through long range interactions activation is mediated through effects on long range organization is not warranted from the data, which are that *Nanog* expression is

correlated with that of PHC1, and disrupting PHC1 disrupts long range interactions. A more in depth analysis of the long range interactions may partially address this—are there enhancers that normally interact with Nanog that no longer interact when PHC1 is knocked out? How does PHC1 binding relate to the long range contacts observed? Do long range contacts correspond to PHC1-only sites, or are they PRC1 sites?

Response: We thank the reviewer for the comments and suggestions. We concur with that the link between PHC1-dependent chromatin organization and gene activation is correlated in our study. We observed that PHC1 interacts with NANOG and *Phc1* deficiency impaired the genomic interactions of the *Nanog* locus in mESCs. Long range gene activation function of Phc1 could be partially achieved through Nanog which promotes chromatin interactions and transcription of pluripotency genes.

To address “are there are enhancers that normally interact with Nanog that no longer interact when PHC1 is knocked out?”, we have now performed Nanog ChIP-PCR on the known enhancer regions bound by Nanog including -45 and +60 SE regions of the *Nanog* locus, *Oct4* and *Phc1* loci in both *Phc1*^{+/+} and *Phc1*^{-/-} mESCs. A non-binding region was used as the negative control. Our results below showed that *Phc1* depletion significantly reduced the occupancy of Nanog on these enhancer regions. Thus, it suggests that Phc1 regulates the binding of Nanog to these sites in mESCs.

In attempt to answer “How does PHC1 binding relate to the long range contacts observed” and “Do long range contacts correspond to PHC1-only sites, or are they PRC1 sites”, we analyzed our 4C-seq data against the published Phc1 and Ring1b ChIP-seq data in mESCs (Kundu et al., *Mol Cell* 2017). Our results below show that there are 3 Phc1-only sites and 8 PRC1 sites (Phc1 and Ring1b co-occupied sites) which interact with the *Nanog* locus and *Phc1* depletion significantly reduced the interactions. These results suggest that PHC1 regulates the chromatin interactions of the *Nanog* locus in both PRC1-dependent and -independent manner in mESCs.

3) The authors demonstrate that Nanog can be co-immunoprecipitated with PHC1, and claim that this interaction is independent of PRC1. The co-IP experiments shown in Fig. 5 provide evidence that Nanog can co-purify with PHC1. However, the evidence that it does not co-purify with PRC1 (e.g. RING1B) is not strong given that there is a faint band in the RING1B IP in 5a, and this experiment was not done in 5c. The RING1B co-IP should also be done in the mESCs, especially since a cross-linking step was used in this case. These blots should be quantified and evidence for reproducibility provided.

Response: We thank the reviewer for the comments and suggestions. To address the reviewer's comments, we have now performed immunoblotting of Ring1b as well as Rybp, subunits of PRC1 in addition to Phc1, after Nanog IP in mESCs. Our results showed that, in contrast to Phc1, Ring1b and Rybp were undetectable in Nanog IP in mESCs. Quantification and reproducibility were also shown below. Therefore, it supports that PHC1 specifically interacts NANOG. These results have been updated in Fig. 5c of the revised manuscript.

4) In their model (Fig. 6), the authors imply that interactions between Nanog and PHC1 recruit PHC1 to change chromatin conformation to activate gene expression. However, no evidence is provided that the interaction between Nanog and PHC1 is involved in regulating transcription. Do Nanog and PHC1 co-localize at genes that are activated when PHC1 is knocked out? A critical test of this model would be to determine whether removing Nanog disrupts targeting of PHC1 to these genes, or conversely whether removing PHC1 disrupts Nanog targeting.

Response: We thank the reviewer for the comments and suggestions. We would like to clarify that we proposed that PHC1 interacts with Nanog to regulate chromatin interactions of the *Nanog* locus and activate pluripotent genes in our model (Fig 6). We observed that PHC1 interacts with NANOG and *Phc1* deficiency impaired the chromatin interactions of the *Nanog* locus in mESCs. This long range gene activation function of Phc1 could be partially and indirectly achieved through Nanog which promotes chromatin interactions and transcription of pluripotency genes.

Our RNA-seq results and analysis of Phc1 and Nanog ChIP-seq data in Fig. 4c showed that many genes for example *Hox* cluster genes in the purple cluster which were upregulated (activated) in *Phc1*^{-/-} mESCs are co-occupied by Phc1 and Nanog. Therefore, it suggests that Phc1 represses differentiation-associated genes of this cluster in a Nanog-dependent manner.

To address “whether removing Nanog disrupts targeting of PHC1 to these genes, or conversely whether removing PHC1 disrupts Nanog targeting” commented by the reviewer, we have now performed Nanog ChIP-PCR on the known binding regions of Nanog including -45 and +60 SE of *Nanog* locus, *Oct4* and *Phc1* in both *Phc1*^{+/+} and *Phc1*^{-/-} mESCs. A non-binding region was used as the negative control. Our results below showed that *Phc1* depletion significantly reduced the occupancy of Nanog on these target regions, supporting that Phc1 regulates the binding of Nanog to the target genes in mESCs.

Additional points:

1) In Figure 1D, is it possible to plot the correlation between the two data sets? What is the basis for concluding that PHC1 and Nanog have similar expression patterns—they do overlap in some cases, but in others (especially later stages, where PHC1 is high but Nanog is low), they do not. This analysis, and the one in 1e would be more compelling if additional Polycomb proteins/genes were analyzed, as in 1a. (i.e. is the relationship between PHC1 and Nanog levels unique?).

Response: We thank the reviewer for the comments and suggestions. The expressions of additional polycomb genes including *RING1B*, *PHC2*, *PHC3*, and *CBX2* have now

been similarly analyzed as *NANOG* and *PHC1* in Fig. 1d. The results below showed that unlike *PHC1* the expressions of these genes are not enriched in EPI lineage in comparison with PE and TE lineages during early human embryonic development. Furthermore, we have now plotted the correlation between *NANOG* with *PHC1* and additional polycomb gene such as *RING1B* which encodes the core subunit of PRC1. The correlation results shown below indicated that the expression of *NANOG* is positively correlated with that of *PHC1*, but not *RING1B*, in the EPI lineage at E5 in particular. Taken together, these results support that *NANOG* specifically associates with *PHC1* at early embryonic developmental stages such as E5. These data have been included as Supplementary Fig. 1 in the revised manuscript.

Fig. S1: Analysis and correlation of PRC1 genes with NANOG expression in early human embryos (Related to Fig. 1). **a**, Analysis of the published single-cell RNA-seq data of early human embryos (E5-7) showing expression of *RING1B*, *PHC2*, *PHC3* and *CBX2* in epiblast (EPI), primitive endoderm (PE) and trophectoderm (TE)³². **b**, Correlation of *NANOG* with *PHC1* and *RING1B* expressions in EPI lineage at E5.

2) The authors state that reducing PHC1 using shRNA does not affect PRC1 levels or activity, based on Fig. 1d. These blots need to be quantified, and evidence for reproducibility shown. It appears that the H2AK119ub1 signal is decreased in the shPHC1.2 cells. Please state how was Nanog quantified—was it normalized to the loading control?

Response: We thank the reviewer for the comments and suggestions. Immunoblotting of NANOG after knockdown of *PHC1* by sh*PHC1.1* and sh*PHC1.2* has now been quantified. Signal intensity was normalized to the loading control ACTIN. Reproducibility data of protein quantification in Fig. 2d were also shown below, $n \geq 3$. Quantification results showed that both sh*PHC1.1* and sh*PHC1.2* significantly decreased the level of Nanog. However, they did not result in consistent decreases in H2AK119ub1 level despite that sh*PHC1.2* significantly reduced it. Moreover, our results demonstrated that they did not significantly change RING1B level and enrichment of H2AK119ub1 on PRC1 target genes as shown by ChIP-PCR analysis (Fig. S3g of the revised manuscript). These results support the conclusion that suppression of *PHC1* did not obviously affect PRC1 activity. We have now rephrased “suppression of *PHC1* expression did not obviously change the levels of RING1B, the E3 ligase subunit of PRC1, and its associated histone modification H2AK119ub1” in the original manuscript to “suppression of *PHC1* expression did not obviously change the levels of RING1B, the E3 ligase subunit of PRC1, and its associated histone modification H2AK119ub1 level was not consistently affected” in the revised manuscript. H2AK119ub1 immunoblotting image was replaced with a new representative panel in Fig. 2d of the revised manuscript as suggested by reviewer 2.

3) How was immunofluorescence quantified and divided into high-low for the graph shown in Fig. 2G? This needs to be detailed in the methods.

Response: We thank the reviewer for the comments and suggestions. ImageJ software was used to quantify the immunofluorescent intensities. The average immunofluorescent intensities of shScr were set as the control. To determine high and low signals, intensities of PHC1 and NANOG staining in shPHC1.1 and shPHC1.2 groups higher or lower than the corresponding controls were defined as PHC1^{high}NANOG^{low} and PHC1^{low}NANOG^{low}, respectively. The percentage of PHC1^{high}NANOG^{low} and PHC1^{low}NANOG^{low} cells were quantified relative to the shScr control. These details have now been included in immunofluorescence in the methods session of the revised manuscript.

4) In Figure 3, analysis of multiple clones should be shown, to confirm that the effects are not due to off-target editing. From the data in Fig. S3, it seems that the correlation between PHC1 knockdown and Nanog levels is not consistent. These data should be quantified. It also needs to be clarified if single clones were used for the experiments, or multiple clones, and if the data were pooled or handled separately.

Response: We thank the reviewer for the comments and suggestions. In our study, we picked 4 *Phc1* knockout mESCs clones and observed that depletion of *Phc1* decreased the expression of Nanog with different extents as shown in both Fig. 3 and Fig. S5a-b (quantification of the data has been included now in Fig. 3b and Fig. S5b of the revised manuscript). We then chose clone 1 which exhibited most significant reduction of Nanog for the rest of the experiments. Furthermore, we overexpressed exogenous *Phc1* in clone 1 and found that it partly rescued Nanog level in *Phc1*^{-/-} mESCs (data was shown below). Taken together, these results preclude CRISPR off-target effect and support that *Phc1* deficiency specifically downregulated Nanog expression in mESCs.

5) Please explain how the colonies were scored for Fig. 3f.

Response: We thank the reviewer for the comment. The details on colony-forming efficiency have now been provided in AP staining in the methods section. To avoid the

ambiguity of undifferentiated, mixed and mixed colonies in Fig. 3f of the original manuscript, we have now scored the total number of colonies 10 days after seeding *Phc1*^{+/+} + vector, *Phc1*^{+/+} + Nanog, *Phc1*^{-/-} + vector, and *Phc1*^{-/-} + Nanog mESCs. These results have now been included in Fig. 3g-h of the revised manuscript.

6) The Nanog rescue experiments (ie Fig. 3e, Fig. 5) should also include a control for overexpression of Nanog in a wild-type background. The level of overexpression of Nanog should be quantified.

Response: We thank the reviewer for the suggestions. We have now overexpressed Nanog in wild-type mESCs and quantified the overexpression level. The overexpression results were included in Fig. 3e-h of the revised manuscript. Quantification of the level of Nanog was provided below.

7) The authors conclude that Oct4, but not Nanog, regulate PHC1 expression. In the Western blot shown in Fig.S4b, it appears that PHC1 levels are increased in the shNanog cells—these results need to be quantified, and evidence for reproducibility shown.

Response: We thank the reviewer for the comments and suggestions. We have now repeated shNanog knockdown and immunoblotting experiments in Fig. S4b of the original manuscript. The results were quantified and included in Fig. S6b of the revised manuscript. Evidence of reproducibility was shown below, n=4.

8) From the methods section, it appears that the 4C-seq experiment was conducted a single time. This experiment needs to be replicated.

Response: We thank the reviewer for the suggestion. We have now replicated 4C-seq experiment and the results were included in Fig 5d-e of the revised manuscript.

9) The methods section of this paper requires additional details. While the different steps involved in many of the protocols are well-described, critical details such as the amounts of reagents used and detailed protocols are not presented (for example, how cells were prepared for immunofluorescence and FACS analysis, how

immunofluorescence experiments were quantified, how many cells were used for Co-IP experiments). Additional details on how the super-resolution microscopy data were collected and analyzed should be provided. The bioinformatics analysis is also not described in sufficient detail—since analysis of publicly available data is critical to the authors argument, exactly how these data we handled and compared should be described (i.e. were peaks called to create Venn diagrams? how were peak calls done? were the data re-analyzed with the same pipeline to make comparisons?).

Response: We thank the reviewer for the comments and suggestions. Additional details for immunofluorescence (last paragraph of page 16 and first paragraph of page 17), FACS analysis (third paragraph of page 17), Co-IP experiments (last paragraph of page 18) have now been presented in the methods section of the revised manuscript. Moreover, more details on the super-resolution microscopy data acquisition and analysis have been provided in the first paragraph in page 17 of the revised manuscript. Besides, additional details on bioinformatics analysis such as peak calling have now been described in the third paragraph in page 20 of the revised manuscript.

Reviewer #2 (Remarks to the Author):

In this manuscript, the authors described that Phc1, a subunit of Polycomb Repressive complex 1 (PRC1), is implicated in Nanog transcriptional activation, which in turn affect pluripotency. The mechanism reported here suggests that Phc1 interacts directly with Nanog to regulate Nanog’s promoter chromatin architecture via stabilization of intra- chromosomal and inter-chromosomal interactions.

Overall this is an interesting manuscript, which unveils a novel PRC1-independent function for Phc1. Nevertheless, several experiments are necessary to fully support the authors’ conclusions, as detailed below.

Main points:

1) To have a complete view of the expression changes of Polycomb subunits during differentiation, the authors should include in figure 1A and 1B the expression levels of PRC2 core components as well as of its associated factors.

Response: We thank the reviewer for the suggestion. We have now examined the expression levels of PRC2 core components including *EZH1*, *EZH2*, *EED* and *SUZ12* in HFFs, hESCs and hiPSCs (see the data below). Our results showed that the expression of *EZH2* in particular was also highly enriched in both hESCs and hiPSCs relative to HFFs. However, its expression did not decline during differentiation of hPSCs. The results have been included in Figs 1a-b of the revised manuscript.

2) Please define the y-axis for Figure 1D. Are the expression levels of Nanog and of PHC1 in the same range?

Response: We thank the reviewer for the suggestion. The y-axis for Fig. 1d has now been defined as RPKM (Reads Per Kilobase of transcript per Million mapped reads) in the revised manuscript. Based on RPKM values, the expression levels of *NANOG* in EPI lineage of E5, E6 and E7 are much higher than that of *PHC1*, whereas *PHC1* transcript is more enriched in EPI lineage of E5, E6 and E7 as opposed to that in PE and TE lineages.

3) The resolution of the figure 2A is too low to appreciate any differences in morphology between WT and KD cells. This should be fixed.

Response: We thank the reviewer for this comment and suggestion. We have now provided images of higher resolution in Fig. 2a of the revised manuscript.

4) At the end of page 5 the authors conclude that “suppression of *PHC1* expression did not obviously change the levels of *RING1B*, the E3 ligase subunit of *PRC1*, and its associated histone modification *H2AK119ub1*”. It seems to me that *H2AK119ub1* levels are indeed lower in *shPHC1.2*. Please comments on this, or provide a more representative western blot panel.

Response: We thank the reviewer for the comment. We concur with that *H2AK119ub1* level appears to be lower in hESCs with *PHC1* knockdown by *shPHC1.2*. As we have responded to Additional Point 2) from Reviewer #1 on pages 10-11 of this letter, our results showed that both *shPHC1.1* and *shPHC1.2* significantly decreased the level of *Nanog*. However, they did not result in consistent decreases in *H2AK119ub1* level despite that *shPHC1.2* reduced it. Moreover, our results demonstrated that they did not significantly change *RING1B* level and enrichment of *H2AK119ub1* on *PRC1* target genes as shown by ChIP-PCR analysis (Fig. S3g of the revised manuscript). These results support the conclusion that suppression of *PHC1* did not obviously affect *PRC1* activity. We have now rephrased “suppression of *PHC1* expression did not obviously change the levels of *RING1B*, the E3 ligase subunit of *PRC1*, and its associated histone modification *H2AK119ub1*” in

the original manuscript to “suppression of *PHC1* expression did not obviously change the levels of RING1B, the E3 ligase subunit of PRC1, and its associated histone modification H2AK119ub1 level was not consistently affected” in the revised manuscript. We have now provided a new representative H2AK119ub1 immunoblotting panel in Fig. 2d of the revised manuscript.

5) It is unclear to me why the level of Nanog are much more effected upon CRISPR KO approach (Figure 2E) when compared to shRNA (Figure 2D), since in both cases the levels of PHC1 downregulation are similar.

Response: We thank the reviewer for the comment. We did observe that the level of NANOG was more affected by CRISPR KO than shRNA knockdown in hESCs. Our explanation of these results is that shRNA knockdown did not affect PRC1 as much as it did for the interaction of PHC1 with NANOG, because most of PHC1 proteins associate with RING1B and only a small fraction of them interact with NANOG in hESCs (Figs S3a-b of the revised manuscript). Therefore, the remaining PHC1 in the knockdown cells was sufficient to maintain the stability of PRC1 (Fig. S3f-g of the revised manuscript), but weaker PHC1-NANOG interaction was likely more sensitive to the reduced level of PHC1 and consequent disruption. By contrast, depletion of *PHC1* by CRISPR in a given cell affected both PRC1 and PHC1-NANOG complex. Loss of PRC1-dependent function of PHC1 caused derepression of developmental genes and impaired pluripotency in addition to the effect of disruption of PHC1-NANOG in the KO cells, which together led to more robust downregulation of NANOG. That's why we chose CRISPR knockout approach in mESCs for the rest of the experiments.

6) The RNA-seq analysis provides important information. Yet this should be integrated with a ChIP-seq data obtained in the same samples. This is even more relevant since the authors relay for ChIP-seq data on a publication which is 15-years old (Boyer et al, 2005).

Response: We thank the reviewer for the comment. We concur with that it would be more relevant to perform ChIP-seq and RNA-seq in the same samples. However, since all the relevant H3K27ac, Nanog, Phc1 and H2AK119ub1 ChIP-seq were previously performed in mESCs and the datasets are publicly available, we chose to analyze these datasets to identify their binding targets and correlate these with our

RNA-seq data. To address the concern of using old polycomb ChIP-seq dataset (we believe this reviewer referred to Boyer et al, Nature 2006; ref 16), we have now analyzed more recent polycomb ChIP-seq dataset in mESCs (Kundu et al., *Mol Cell* 2017). These results have now been updated in Fig. 4 of the revised manuscript.

7) It is important that the interaction between Nanog and Phc1 does not involve other subunits of the PRC1 complex. For this an IP for Nanog should be performed followed by Western blot for PRC1 subunits.

Response: We thank the reviewer for the comment and suggestion. We have now

performed immunoblotting of Ring1b as well as Rybp, subunits of PRC1 in addition to Phc1, after Nanog IP in mESCs. Our results showed that, in contrast to Phc1, Ring1b and Rybp were undetectable in Nanog IP in mESCs. Thus, it supports that PHC1 specifically interacts NANOG. These results have been updated in Fig. 5c of the revised manuscript.

REVIEWER COMMENTS

Reviewer #1 (Remarks to the Author):

This is a carefully crafted revision in which the authors have systematically addressed the concerns that were raised. The one remaining concern I have is with the immunoprecipitation data in Figure 5. While the quality of the data has been improved, I think the full blots should be shown (perhaps as source data), and that the images used in the figure should be cropped less extensively. It is disconcerting to have partially cut off bands that are clearly enriched in the IPs—for example the anti-Flag blot for the HA IP in the bottom panel of Fig. 5B. If there are extra bands present in the IPs (as is clearly the case for most of the blots), I think these should be shown, and the authors can simply indicate which bands are correct.

Reviewer #2 (Remarks to the Author):

The authors have taken in consideration all the criticisms previously raised by this referee. The new data confirm and extend the original findings.

Reviewer #3 (Remarks to the Author):

As requested by the editor, my comments only focus on the 4C / Hi-C data analysis, which was used in this study to show the chromatin interaction dynamics of the Nanog locus between Phc1+/+ and Phc1-/- mESC cells. Overall, I think the quality of the 4C data generated by this study is not good enough to support its conclusions.

1. As reviewer #1 pointed out, it's crucial to show that the individual interaction changes are reproducible between biological replicates of 4C-Seq. Although the authors replicated the 4C-Seq experiments in the revised manuscript, for analysis, they are still calculating the average contact profile of the replicates, therefore, it's not clear whether the decreased contacts with Apobec1, Slc2a3, Klf5 and Sall1 are reproducible between replicates or not.
2. From the contact profile showed in current figure 5d (the green bar), I don't see significant change at the -45kb SE locus as described by the authors.
3. I highly recommend the authors perform a thorough quality assessment of their 4C data and provide the key metrics for both replicates, such as the fragMappedCisPercCorr (percentage of reads mapped to the viewpoint chromosome excluding reads mapped to the top fragment ends, need to be >60%) and the capt100Kb (percentage of unique fragment ends within 100Kb of the viewpoint with at least 1 mapped read, need to be >40%).
4. The resolution of Hi-C data selected by this study is not high enough to give any meaningful information at the Nanog locus. I suggest using the more recent high-resolution in situ Hi-C data generated by Bonev 2017.
5. The KR-normalized contact values cannot be directly compared between Hi-C experiments, therefore, the differences between mESC and Cortex showed in Fig. S7 are likely from the differences between sequencing depths.
6. Different reference genomes were used in mapping 4C and Hi-C reads.

Reference:

1. Krijger, P. H. L., Geeven, G., Bianchi, V., Hilvering, C. R. E. & de Laat, W. 4C-seq from beginning to end: A detailed protocol for sample preparation and data analysis. *Methods* 170, 17-32,

doi:10.1016/j.ymeth.2019.07.014 (2020).

2. Bonev, B. et al. Multiscale 3D Genome Rewiring during Mouse Neural Development. *Cell* 171, 557-572 e524, doi:10.1016/j.cell.2017.09.043 (2017).

Reviewers' comments:

Reviewer #1 (Remarks to the Author):

This is a carefully crafted revision in which the authors have systematically addressed the concerns that were raised. The one remaining concern I have is with the immunoprecipitation data in Figure 5. While the quality of the data has been improved, I think the full blots should be shown (perhaps as source data), and that the images used in the figure should be cropped less extensively. It is disconcerting to have partially cut off bands that are clearly enriched in the IPs—for example the anti-Flag blot for the HA IP in the bottom panel of Fig. 5B. If there are extra bands present in the IPs (as is clearly the case for most of the blots), I think these should be shown, and the authors can simply indicate which bands are correct.

Response: We thank the reviewer for the excellent comments and suggestions. To address the concern on cropping IP images, the full blots for the IP images in Fig. 5b were shown below and improved IP images were provided in the bottom panel of Fig. 5b in the revised manuscript.

Upper left: the full blots from which we cropped the bands (rectangle) for images of anti-FLAG immunoblotting after IP FLAG in the upper panel of Fig. 5b of the revised manuscript. Due to differences in FLAG signal intensity between input and IP FLAG, images of long and short time exposure were used for input and IP FLAG, respectively. Lanes 1, 4 and 7 from left to right are protein ladders.

Upper right: the full blots from which we cropped the bands (rectangle) for images of anti-HA immunoblotting after IP FLAG in the upper panel of Fig. 5b.

Bottom left: the full blots from which we cropped (rectangle) for images of anti-FLAG immunoblotting after IP HA in the bottom panel of Fig.5b.

Bottom right: the full blots from which we cropped (rectangle) for images of anti-HA immunoblotting after IP HA in the bottom panel of Fig.5b.

The full blots will be included as source data as the reviewer suggested. We have now improved IP and provided better quality of images in the bottom panel of Fig. 5b as shown below in the revised manuscript.

Reviewer #2 (Remarks to the Author):

The authors have taken in consideration all the criticisms previously raised by this referee. The new data confirm and extend the original findings.

Response: We thank the reviewer for the comments.

Reviewer #3 (Remarks to the Author):

As requested by the editor, my comments only focus on the 4C / Hi-C data analysis, which was used in this study to show the chromatin interaction dynamics of the Nanog locus between Phc1^{+/+} and Phc1^{-/-} mESC cells. Overall, I think the quality of the 4C data generated by this study is not good enough to support its conclusions.

1. As reviewer #1 pointed out, it's crucial to show that the individual interaction changes are reproducible between biological replicates of 4C-Seq. Although the authors replicated the 4C-Seq experiments in the revised manuscript, for analysis, they are still calculating the average contact profile of the replicates, therefore, it's not clear whether the decreased contacts with Apobec1, Slc2a3, Klf5 and Sall1 are reproducible between replicates or not.

Response: We thank the reviewer for the excellent suggestions and comments. We concur that it was not clear to evaluate the reproducibility of the decreased contacts of *Nanog* promoter (anchor) with Apobec1, Slc2a3, Klf5 and Sall1 loci and their neighboring regions by showing the average contact profile of the two replicates in Fig 5d. To address the reproducibility issue, we have now performed another two

replicates of 4C-seq and displayed each contact profile of the four replicates as Rep1, Rep2, Rep3 and Rep4 in Fig. S7a. Despite variations in the contact reads between each replicate, we observed consistently decreased contacts of the *Nanog* promoter with regions near its -45kb super-enhancer (SE) and *Slc2a3* loci in *Phc1*^{-/-} than *Phc1*^{+/+} mESCs across our four replicates of 4C-seq datasets (Fig. S7a). Moreover, the decreased contacts at these regions were also observed in *Nanog* knockdown mESCs compared to the control after we analyzed the previously published 4C-seq data (de Wit et al., Nature 2013) (Fig. 5d Middle panel). Statistical analysis demonstrated that *Phc1* deficiency (*Phc1*^{-/-}) significantly decreased the interactions of the *Nanog* promoter with regions near its -45kb SE and *Slc2a3* loci in chr6 (Fig. 5d Top panel). Nevertheless, we did not observe consistent contact changes of the *Nanog* promoter with *Apobec1* in chr6, *Sall1* in chr8 and *Klf5* in chr14 between *Phc1*^{+/+} and *Phc1*^{-/-} mESCs across four replicates (data shown below). In line with this, analysis of the published 4C-seq datasets showed that *Nanog* knockdown did not lead to decrease in contacts of the *Nanog* promoter with these loci in the genome (de Wit et al., Nature 2013) (data shown below). This inconsistency was likely due to the overall low contacts of these loci (*Sall1* and *Klf4*) with the *Nanog* promoter as seen from our 4C-seq datasets and the previously published 4C-seq data (data shown below). However, analysis of our four replicates of 4C-seq datasets showed consistent changes in other inter-chromosomal interactions with the *Nanog* promoter between *Phc1*^{+/+} and *Phc1*^{-/-} mESCs (Fig. S7b, Supplementary Table 8). Therefore, our results overall support the conclusion that PHC1 is involved in organizing chromatin interactions of the *Nanog* locus partially through interacting with NANOG. Thus, we have now included interaction profiles of regions containing -45kb SE and *Slc2a3* in chr6 from four replicates of 4C-seq in Fig. S7a and removed *Sall1* and *Klf5* contact profiles in Fig. 5d of the revised manuscript. Circos plot showing consistent changes in genome-wide interactions with the *Nanog* locus from four replicates of 4C-seq of *Phc1*^{+/+} and *Phc1*^{-/-} mESCs has now been included as Fig S7b in the revised manuscript. The chromosomal regions with consistent changes in the interactions with the *Nanog* locus between *Phc1*^{+/+} and *Phc1*^{-/-} mESCs were provided in Supplementary table 8 in the revised manuscript. Normalized interaction intensity profile of our four 4C-seq datasets showing regions near -45kb SE and *Slc2a3* with significant decreases in contacts with the *Nanog* promoter in chr6 of *Phc1*^{-/-} than *Phc1*^{+/+} mESCs (marked by asterisk) has been included in top panel of Fig 5d in the revised manuscript. The text and legends associated with changes in Fig 5d and Fig S7 have been revised accordingly in the manuscript.

Fig. S7: a, Analysis of four replicates of 4C-seq datasets showing interaction profiles of genomic regions with the *Nanog* promoter (anchor) in chr6 of *Phc1*^{+/+} (green line) and *Phc1*^{-/-} (purple line) mESCs. X and Y axes indicate chromosome coordinates around the *Nanog* locus and the normalized interaction intensity, respectively. Chr6: 122500000-122700000 of mm9 genome assembly were shown. The peak interaction was normalized to the highest interaction fragment (chr6: 122654169 fragment) in the control sample of each replicate. **b**, Circos plot of inter- and intra-chromosomal interactions with the *Nanog* promoter (anchor) in chr6 of mm9 genome assembly in *Phc1*^{+/+} (green line) and *Phc1*^{-/-} (purple line) mESCs, respectively (n=4). **c**, Hi-C interaction matrices of mouse CN and mESCs in chr6: 122500000-122700000 of mm9 genome, respectively (Bonev et al., Cell 2017). **d**, Hi-C interaction matrices of mouse CN and mESCs in an extended region in chr6: 120049982-125249982 of mm9 genome assembly, respectively (Bonev et al., Cell 2017).

Fig. 5d: Top panel: Average interaction profile showing the normalized interaction intensity (y axis) of regions with the *Nanog* promoter (anchor) in chr6: 122500000-122700000 of mm9 genome assembly (x axis) in *Phc1*^{+/+} (green line) and *Phc1*^{-/-} (purple line) mESCs. Regions with significant decreases in *Phc1*^{-/-} than *Phc1*^{+/+} mESCs were highlighted and statistical significance was indicated with asterisk. T-tests were used. * $P < 0.05$, $n = 4$. **Middle panel:** Analysis of the previously published 4C-seq of genomic regions near the *Nanog* promoter in chr6 of control and *Nanog* knockdown (KD) mESCs (de Wit et al., Nature 2013); H3K27ac and *Nanog* ChIP-seq binding profiles in the corresponding regions. **Bottom panel:** Hi-C interaction matrices of mouse cortical neurons (CN) and mESCs in chr6: 122500000-122700000 of mm9 genome assembly, respectively (Bonev et al., Cell 2017) (Hi-C contact matrix at 5kb resolution).

Top panel: Analysis of four replicates of 4C-seq showing interaction profiles of Sall1 in chr8 and Klf5 in chr14 with the *Nanog* promoter (anchor) in *Phc1*^{+/+} (green line) and *Phc1*^{-/-} (purple line) mESCs. **Middle panel:** Analysis of the previously published 4C-seq of Sall1 in chr8 and Klf5 in chr14 interactions with the *Nanog* promoter (anchor) of control and Nanog knockdown (KD) mESCs (de Wit et al., Nature 2013). **Bottom panel:** H3K27ac and Nanog ChIP-seq binding profiles at the corresponding regions.

2. From the contact profile showed in current figure 5d (the green bar), I don't see significant change at the -45kb SE locus as described by the authors.

Response: We thank the reviewer for the excellent comment. As we have explained in our response to the first comment from this reviewer above, we have now performed statistical analysis of our four replicates of 4C-seq datasets and the results showed that the interactions of regions near -45kb SE and Slc2a3 with the *Nanog* promoter were significantly decreased in *Phc1*^{-/-} than *Phc1*^{+/+} mESCs (* $p < 0.05$, $n = 4$). The normalized interaction intensity of our four 4C-seq datasets showing regions near -45kb SE and Slc2a3 with significant decreases in contacts with the *Nanog* promoter in chr6 (highlighted and marked by asterisk) were included as Fig 5d (Top panel) in the revised manuscript.

Fig. 5d: Top panel: Average interaction profile showing the normalized interaction intensity (y axis) of regions with the *Nanog* promoter (anchor) in chr6: 122500000-122700000 of mm9 genome assembly (x axis) in *Phc1*^{+/+} (green line) and *Phc1*^{-/-} (purple line) mESCs. Regions with significant decreases in *Phc1*^{-/-} than *Phc1*^{+/+} mESCs were highlighted and statistical significance was indicated with asterisk. T-tests were used. * $P < 0.05$, $n = 4$. **Middle panel:** Analysis of the previously published 4C-seq of genomic regions near the *Nanog* promoter in chr6 of control and Nanog knockdown (KD) mESCs (de Wit et al., Nature 2013); H3K27ac and Nanog ChIP-seq binding profiles in the corresponding regions. **Bottom panel:** Hi-C interaction matrices of mouse cortical neurons (CN) and mESCs in chr6: 122500000-122700000 of mm9 genome assembly, respectively (Bonev et al., Cell 2017) (Hi-C contact matrix at 5kb resolution).

3. I highly recommend the authors perform a thorough quality assessment of their 4C data and provide the key metrics for both replicates, such as the fragMappedCisPercCorr (percentage of reads mapped to the viewpoint chromosome excluding reads mapped to the top fragment ends, need to be >60%) and the capt100Kb (percentage of unique fragment ends within 100Kb of the viewpoint with at least 1 mapped read, need to be >40%).

Response: We thank the reviewer for the excellent suggestions and comments. We have now performed the quality assessment of our four replicates of 4C-seq datasets and provided the key metrics in the table below. Although the fragMappedCisPercCorr of our 4C data were slightly lower than 60% recommended by this reviewer, consistent decreases in the interactions of regions near -45kb SE and

Slc2a3 loci with the *Nanog* promoter were observed in *Phc1*^{-/-} mESCs compared to *Phc1*^{+/+} mESCs across our four replicates of 4C-seq datasets. Moreover, the contact changes at these regions were also observed in the *Nanog* KD mESCs in comparison with the control after analysis of the previously published 4C-seq (de Wit et al., Nature 2013). Therefore, we think that these results support our overall conclusion that PHC1 is involved in organizing chromatin interactions of the *Nanog* locus partially through interacting with NANOG in the manuscript.

Sample Id	Total number of reads	Reads in cis	Percentage of reads in cis
Phc1 ^{+/+} _Rep1	13547138	6866571	0.506865066
Phc1 ^{+/+} _Rep2	7491564	3126112	0.417284295
Phc1 ^{+/+} _Rep3	17207650	8522577	0.495278379
Phc1 ^{+/+} _Rep4	8847673	3663198	0.414029542
Phc1 ^{-/-} _Rep1	13730760	7956904	0.579494799
Phc1 ^{-/-} _Rep2	9026866	3786696	0.41949177
Phc1 ^{-/-} _Rep3	13571106	7517200	0.553912113
Phc1 ^{-/-} _Rep4	8504155	3882014	0.456484389

4. The resolution of Hi-C data selected by this study is not high enough to give any meaningful information at the *Nanog* locus. I suggest using the more recent high-resolution in situ Hi-C data generated by Bonev 2017.

Response: We thank the reviewer for the excellent comment and suggestion. We have now analyzed high-resolution Hi-C data generated by Bonev et al (Cell 2017) as the reviewer recommended. The new data below showed that there was notable formation of TAD boundaries of the *Nanog* locus in mESCs in contrast to mouse cortical neurons (CN). These new high-resolution results have now been included in the bottom panel of Fig. 5d in the revised manuscript.

(Bonev et al., Cell 2017)

Hi-C interaction matrices of mouse cortical neurons (CN) and mESCs in chr6: 122500000-122700000 of mm9 genome, respectively (Hi-C contact matrix at 5kb resolution).

In addition, we also re-analyzed the Hi-C data generated by Dixon et al (Nature 2012) with improved resolution (5kb) and the data below also showed that there were increased interactions of the *Nanog* locus in mESCs in comparison with that in mouse cortex. Therefore, results from these two datasets together support that at the three-dimensional genome level, regions of the *Nanog* locus exhibit increased interactions in mESCs as opposed to the differentiated cells.

(Dixon et al., Nature 2012)

Hi-C interaction matrices of mouse cortex and mESCs in chr6: 122500000-122700000 of mm9 genome, respectively (Hi-C contact matrix at 5-kb resolution).

5. The KR-normalized contact values cannot be directly compared between Hi-C experiments, therefore, the differences between mESC and Cortex showed in Fig. S7 are likely from the differences between sequencing depths.

Response: We thank the reviewer for the excellent comments. To exclude the confounding effect of variable sequencing depths on the comparison of the analyzed data, we randomly sampled equal numbers of read pairs from each condition for downstream analyses involving comparison between conditions. By doing so, Hi-C data from these two independent studies (Dixon et al., *Nature* 2012; Bonev et al., *Cell* 2017) still consistently showed increased genome-wide chromatin interactions in mESCs compared to CN as shown below. Therefore, the differences between mESCs and Cortex/CN were unlikely due to the effect of sequencing depths. The new analysis results of Hi-C data generated by Bonev 2017 have now been included in Fig. S7c-d in the revised manuscript.

Top panel: Hi-C interaction matrices of mouse cortical neurons (CN) and mESCs in chr6: 122500000-122700000 of mm9 genome, respectively. Bottom panel: Hi-C interaction matrices of mouse cortical neurons (CN) and mESCs in an extended region chr6: 120000000-125200000 of mm9 genome, respectively (Hi-C contact matrix in 5-kb resolution).

(Dixon et al., Nature 2012)

Top: Hi-C interaction matrices of mouse cortex and mESCs in chr6: 122500000-122700000 of mm9 genome, respectively. Bottom: Hi-C interaction matrices of mouse cortex and mESCs in an extended region chr6: 120000000-125200000 of mm9 genome, respectively (Hi-C contact matrix in 5-kb resolution).

6. Different reference genomes were used in mapping 4C and Hi-C reads.

Reference:

1. Krijger, P. H. L., Geeven, G., Bianchi, V., Hilvering, C. R. E. & de Laat, W. 4C-seq from beginning to end: A detailed protocol for sample preparation and data analysis. *Methods* 170, 17-32, doi:10.1016/j.ymeth.2019.07.014 (2020).
2. Bonev, B. et al. Multiscale 3D Genome Rewiring during Mouse Neural Development. *Cell* 171, 557-572 e524, doi:10.1016/j.cell.2017.09.043 (2017).

Response: We thank the reviewer for the comment. We have now used genome mm9 as the reference genome for the analysis of both 4C-seq and Hi-C data in the revised manuscript.

REVIEWER COMMENTS

Reviewer #3 (Remarks to the Author):

The authors have addressed all my previous concerns. However, I have one more concern regarding the analysis of the published 4C-Seq data (de Wit E et al., Nature 2013): we usually expect that the highest signals happen at the bait region in a 4C experiment, why signals are depleted at the Nanog bait region in Fig. 5d (middle panel)? Importantly, the calculated 4C contact profiles of both control and Nanog KD cells in Fig. 5d are dramatically different from what Supplementary Figure 15 (see the attachment) shows in the original publication (de Wit E et al., Nature 2013). To me, the whole analysis might be wrong or the authors missed important details in the method section.

Reviewer #3 (Remarks to the Author):

The authors have addressed all my previous concerns. However, I have one more concern regarding the analysis of the published 4C-Seq data (de Wit E et al., Nature 2013): we usually expect that the highest signals happen at the bait region in a 4C experiment, why signals are depleted at the Nanog bait region in Fig. 5d (middle panel)? Importantly, the calculated 4C contact profiles of both control and Nanog KD cells in Fig. 5d are dramatically different from what Supplementary Figure 15 (see the attachment) shows in the original publication (de Wit E et al., Nature 2013). To me, the whole analysis might be wrong or the authors missed important details in the method section.

Response: We thank the reviewer for the excellent comment. We regret that we did make a mistake as the reviewer pointed out. However, we would like to clarify that, when we analyzed the published 4C-seq datasets (de Wit E et al., Nature 2013), we realized that the primers for the viewpoints of the examined genes including *Nanog* gene locus were not provided in the paper. After aligning and obtaining the coverage of each fragment in the 4C-seq datasets, we noticed that two fragments near the *Nanog* promoter region had the highest reads coverage shown as Fragment A and C in Figure 1 below (thin blue and brown lines represent reads on these two Fragments, respectively). During our initial analysis, we thought these abundant reads on these two fragments were due to undigested and self-ligated fragments which is usually the case and therefore should be removed before subsequent analysis. When we did so, it led to the depletion of signals near the viewpoints as the reviewer pointed out in Fig. 5d Middle panel in our manuscript. However, when we took a closer look at the published 4C-seq datasets in an attempt to address the reviewer's comments, we now realized that there are three Hind III cut sites in the *Nanog* promoter region and digestion with Hind III which was used for the 4C experiments in their paper should have generated four fragments (A, B, C and D). Given the circumstance with the lack of *Nanog* viewpoint primer information even after repeated attempts to request by contacting the correspondence author of the paper, we can only speculate that the viewpoint primers are likely within the left end of Fragment D close to Hind III cut site (red arrow in Figure 1), because Fragment D contains the TSS and promoter region of *Nanog* gene (about 120bp before TSS). Such fragments are typically used as bait regions to explore the interaction between gene promoter and other *cis* elements. Therefore, what we suppose to delete are the high reads at the right ends of Fragments C and D due to the undigested and self-ligated fragments during 4C experiment respectively. We would like to point out that the comparison of the raw data without removal of these reads also showed the decrease in the contact intensities at -45kb SE in particular with the *Nanog* promoter in the Nanog knockdown mESCs compared to the control in Figure 2 below (the red and yellow boxes show the *Nanog* promoter and -45kb SE regions, respectively). These results suggest that change in the analysis would not affect the observed differences between Nanog KD and control mESCs.

Figure 1. 4C raw data diagram

Figure 2. 4C raw data around the *Nanog* locus shown in IGV browser
 (The light blue arrow indicates the fragment with high signals which is located near -45kb SE region in control and *Nanog* KD mESCs.)

Therefore, we re-processed the published 4C-seq datasets (de Wit E et al., Nature 2013) as described in our manuscript after removal of the reads at the right ends of fragments C and D likely due to undigested and self-ligated fragments. This led to the appearance of expected high signals near the empirical viewpoint as shown in Figure 3 below (highlighted panel). Both raw data (Figure 2) and normalized data (highlighted panel in Figure 3 below) showed that the interaction intensities at -45kb SE decreased in *Nanog* knockdown mESCs in comparison with the control. This

result has now been included in Fig. 5d Middle panel in the revised manuscript.

Figure 3. 4C-seq analysis around the *Nanog* locus in Fig. 5d

Regarding the reviewer’s second comment on “Importantly, the calculated 4C contact profiles of both control and *Nanog* KD cells in Fig. 5d are dramatically different from what Supplementary Figure 15 (see the attachment) shows in the original publication (de Wit E et al., Nature 2013).”, we would like to point out that this seemingly discrepancy likely arose from lack of the abscissa unit in Figure S15. The coordinate for Figure S15b was very likely mb as what has been shown in the main figures in the original paper. Therefore, it was the comparison between *Nanog* KD and control mESCs on the whole chromosome 6 rather than regions near the *Nanog* promoter. Moreover, our analysis of their 4C-seq datasets for control and *Nanog* KD mESCs on whole chromosome 6 also exhibited very similar profiles to that in Fig. S15b of the original paper as shown in Figure 4 below. Therefore, our analysis was overall consistent with the results in the original paper and did not affect the observed decreases in the interaction intensities of the *Nanog* promoter with regions at -45kb SE in particular in *Nanog* KD mESCs compared to control.

de Wit E et al., Nature 2013. Fig S15b

4C raw data processed by us

Figure 4. 4C signals on chromosome 6

REVIEWERS' COMMENTS

Reviewer #3 (Remarks to the Author):

The authors have thoroughly answered my questions, I appreciated the detailed explanation and have no more concerns.